# Dislocation interactions in olivine control postseismic creep of the upper mantle

David Wallis [1,5✉], Lars N. Hansen[2], Angus J. Wilkinson [3] & Ricardo A. Lebensohn [4]

Changes in stress applied to mantle rocks, such as those imposed by earthquakes, commonly induce a period of transient creep, which is often modelled based on stress transfer among slip systems due to grain interactions. However, recent experiments have demonstrated that the accumulation of stresses among dislocations is the dominant cause of strain hardening in olivine at temperatures ≤600 °C, raising the question of whether the same process contributes to transient creep at higher temperatures. Here, we demonstrate that olivine samples deformed at 25 °C or 1150–1250 °C both preserve stress heterogeneities of ~1 GPa that are imparted by dislocations and have correlation lengths of ~1 μm. The similar stress distributions formed at these different temperatures indicate that accumulation of stresses among dislocations also provides a contribution to transient creep at high temperatures. The results motivate a new generation of models that capture these intragranular processes and may refine predictions of evolving mantle viscosity over the earthquake cycle.

---

[1] Department of Earth Sciences, Utrecht University, Utrecht, The Netherlands. [2] Department of Earth and Environmental Sciences, University of Minnesota-Twin Cities, Minneapolis, MN, USA. [3] Department of Materials, University of Oxford, Oxford, UK. [4] Los Alamos National Laboratory, Los Alamos, NM, USA. [5] Present address: Department of Earth Sciences, University of Cambridge, Cambridge, UK. ✉email: dw584@cam.ac.uk

Major earthquakes impose changes in stress on the hot, viscous rocks underlying the fault zone[1–7]. Relaxation of these stresses contributes to postseismic deformation and stress redistribution within and between fault zones over the earthquake cycle[1–7]. Modelling these processes is challenging because changes in the stress applied to viscous rocks induce a period of evolution in viscosity, which is observable in laboratory experiments[8–14] and detectable in geodetic datasets[1–4,15]. In both these contexts, the observations are often best fit by rheological models with non-linear stress dependencies indicating that dislocation motion plays a dominant role[1–4,8–10]. Experiments on geological materials[9,16–18] and metals[19,20] have demonstrated that this non-linear transient creep results from subtle changes in the dislocation microstructure and/or micromechanical state in response to a change in applied stress. Constraints on these underlying microphysical processes are therefore key to the formulation of rheological models that can be confidently extrapolated from laboratory conditions to those of natural deformation.

Previous studies have considered rheological behaviours typical of both low-temperature and high-temperature conditions (the transition from low to high corresponding very approximately to 50% of the melting temperature) to contribute to postseismic transient creep. Geological studies have focused on deformation microstructures in exhumed shear zones that are inferred to have directly underlain frictional faults, in which the stress changes and their associated microstructural expressions are most pronounced[6,21]. In these settings, postseismic deformation is recorded by dislocation microstructures (e.g., slip bands and cells of tangled dislocations) associated with low-temperature plasticity that is inferred to have been induced by transient increases in stress[6,21]. In contrast, postseismic geodetic signals can capture far-field deformation distributed throughout the lower crust and upper mantle, typically at greater depths where temperatures are higher and stress changes are smaller than directly adjacent to the seismogenic zone[1–4,15,22]. In these portions of the lithosphere and the asthenosphere, deformation by dislocation creep or dislocation-accommodated grain boundary sliding is likely commonplace[23–25].

Most quantitative descriptions of transient creep used to analyse postseismic geodetic data are largely phenomenological in that they describe observed material behaviour but lack a comprehensive basis in the underlying physical processes. A widely used example is the Burgers model[1,2,26–28], which is capable of generating a time-dependent evolution in viscosity, anelastic behaviour and steady-state flow. Whilst attempts have been made to link a Burgers model to transient dislocation creep[1,2,28], the overarching arrangement of the Maxwell and Kelvin elements composing a Burgers model does not arise naturally and exclusively from the fundamentals of dislocation motion[26].

A model with a basis in the microphysics of dislocation creep has been proposed by Karato[29], inspired by earlier experiments on water ice[12,13]. In this model, transient creep arises from stress transfer between grains with different crystallographic orientations. Upon loading, initial dislocation motion occurs in grains with high resolved shear stress on the weakest slip system. However, maintenance of strain compatibility in an aggregate of anisotropic grains requires activation of additional, stronger slip systems in other grains to counteract stress concentrations generated during progressive deformation. This progressive transition in the slip systems that control the bulk strain rate results in a progressive increase in viscosity[29].

Whilst the transfer of stress among slip systems in grains of different orientations does potentially contribute to transient creep of rocks, a variety of studies have demonstrated that transient creep also occurs even in single crystals, for which the intergranular stress-transfer model cannot apply. Strain-hardening transients are exhibited by single crystals of olivine deforming by both low-temperature plasticity[30–32] and power-law creep at high temperatures[9,33,34], with the transition between mechanisms occurring at temperatures of ~1000–1100 °C at typical experimental stresses and strain rates. Microstructural analyses of single crystals deformed in both temperature regimes indicate that strain hardening results from increases in dislocation density[9,35–37], short-range dislocation interactions[33], and long-range interactions among dislocations via their stress fields[37,38]. Short-range interactions involve local rearrangement of atoms in the vicinity of adjacent dislocation cores. Long-range interactions occur when the stress fields of nearby dislocations do not cancel so that a dislocation may exert forces on its neighbours and those further afield without the dislocation cores coming into contact. These observations of single crystals imply that these intragranular processes likely also make an essential contribution to strain hardening of aggregates that has been largely overlooked in models formulated for geodetic studies[1,2,29]. This suggestion is supported by the practically indistinguishable shapes of the stress–strain curves of single crystals and aggregates of olivine between the yield stress and the flow stress when deformed at room temperature[31]. The behaviour of both the single crystals and aggregates can be quantitatively described by a single model based on long-range dislocation interactions[31]. Therefore, it seems clear that transient creep of aggregates deforming by low-temperature plasticity is primarily controlled by intragranular dislocation interactions. However, a major outstanding question remains. Do dislocation interactions contribute significantly to transient deformation at the high temperatures relevant to the regions of the lithosphere and asthenosphere contributing to postseismic creep?

We hypothesise that the same intragranular processes control transient creep of aggregates of olivine deformed by power-law creep and those deformed by low-temperature plasticity. We suggest that samples deformed in either regime share similarities in characteristics of their residual stress fields. Specifically, we test for the presence of intragranular stress heterogeneity, whether that heterogeneity can be attributed to the dislocation content, and whether that heterogeneity occurs over length-scales greater than the average dislocation spacing, indicating that the stresses arise from long-range dislocation interaction.

We analyse intragranular stress heterogeneity within olivine deformed at either low temperature and high confining pressure (25 °C, 6.9–9.3 GPa)[31] or high temperature and low confining pressure (1150–1250 °C, 300 MPa)[23]. We analyse these samples with high-angular resolution electron backscatter diffraction (HR-EBSD) (Methods), which maps lattice distortion by using cross-correlation to track shifts in subregions within diffraction patterns. Unlike conventional EBSD, which struggles to resolve the subtle microstructural changes associated with transient creep at small strains[39], HR-EBSD provides exceptionally precise estimates of the density of geometrically necessary dislocations (GNDs, the fraction of the total dislocation density that generates net lattice curvature and long-range stress heterogeneity) and, importantly, maps heterogeneity in elastic strain and residual stress stored in the samples after the experiments[40–43]. We analyse the stress distributions in terms of the theory, established in the materials sciences[44–47], for stress fields of a population of dislocations to test the causality between stress heterogeneity and the dislocation content (Methods). In particular, we test whether tails of the probability ($P$) distributions of shear stress ($\sigma_{12}$) follow a $P(\sigma_{12}) \propto |\sigma_{12}|^{-3}$ relationship, as expected of stress fields generated by dislocations[44–47] (Methods). Autocorrelation of the stress fields provides a measure of the characteristic length scale of stress variation (Methods) and therefore a test for the presence

of long-range internal stress, which varies over length scales greater than the average dislocation spacing.

The HR-EBSD data reveal striking similarities in GND densities and stress heterogeneity between the samples deformed at low and high temperatures. Both sets of samples contain GND densities on the order of $10^{14}$ m$^{-2}$ and stress heterogeneity on the order of 1 GPa. Importantly, the probability distributions of stress heterogeneity do follow a $P(\sigma_{12}) \propto |\sigma_{12}|^{-3}$ relationship at high stresses, confirming that the stress heterogeneity is imparted by dislocations. Moreover, the correlation lengths of the stress fields are on the order of 1 μm in both sets of samples, demonstrating the presence of long-range internal stress and the occurrence of long-range dislocation interactions. These observations provide the basis for a new generation of rheological models of transient creep that incorporate the microphysics of intracrystalline deformation.

## Results

**Geometrically necessary dislocation density**. Figure 1a presents maps of the densities of GNDs in each sample (Tables 1 and 2, Methods). The undeformed single crystal, MN1, exhibits an apparent GND density of ~$10^{12}$ m$^{-2}$, resulting from noise in the rotation measurements[48]. This value is consistent with a total dislocation density of <$10^{10}$ m$^{-2}$ measured previously by oxidation decoration[48]. The undeformed aggregate, PI-1523s, was synthesised by hot isostatic pressing at a temperature of 1200 °C. Some grains within this sample are dissected by bands of GND densities >$10^{14}$ m$^{-2}$ that mark subgrain boundaries. However, many grains lack subgrain boundaries. Densities of GNDs within subgrain interiors and within grains that lack subgrain boundaries are homogeneous and typically ≲$10^{13.5}$ m$^{-2}$, corresponding to the noise floor of the measurements (Methods). The single crystal deformed at room temperature, San382t, displays parallel linear arrays of elevated GND density, on the order of $10^{14}$ m$^{-2}$. In contrast, the aggregates deformed at room temperature, San382b and San372b, lack this linear structure and contain few regions with GND densities <$10^{14}$ m$^{-2}$. Instead, the grains in these samples contain irregular-shaped patches, a few micrometres across, with GND densities in the range $10^{14}$–$10^{15}$ m$^{-2}$. The samples deformed at high temperatures, PI-1488, PI-1523 and PI-1519, contain GND densities broadly comparable to those of San382b but typically exhibit smoother variations in GND density within grains and more linear arrays of GNDs (e.g., PI-1523) than the aggregates deformed at room temperature.

**Spatial heterogeneity in residual stress**. Figure 1b presents maps of $\sigma_{12}$ normalised by subtracting the mean value within each grain (Methods). The single-crystal control sample, MN1, exhibits a homogeneous stress distribution with a standard deviation of 70 MPa. The isostatically hot-pressed aggregate, PI-1523s, exhibits stress heterogeneity that typically varies smoothly by a few 100 MPa within grains, along with occasional stress concentrations. All other samples exhibit more pronounced intragranular stress heterogeneity. Sample San382t contains bands of elevated stress of alternating sign that vary in magnitude on the order of 1 GPa over distances of a few micrometres. The deformed aggregates, irrespective of the temperature at which they were deformed, exhibit spatial distributions of stress heterogeneity that are qualitatively similar to each other and lack the ordered structure displayed by San382t. In each case, stress typically varies smoothly between domains of stress on the order of 1 μm across, with alternating sign and magnitudes again on the order of 1 GPa. Little anisotropy is evident within individual grains in the aggregates.

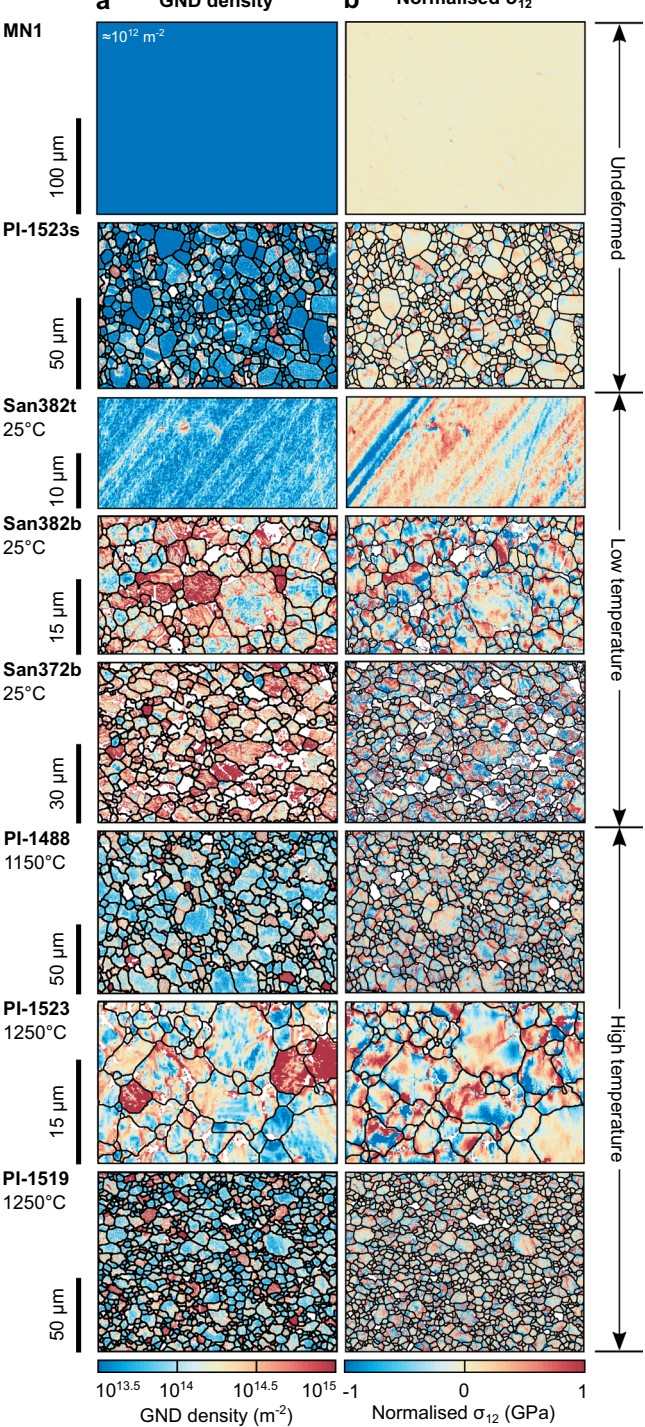

**Fig. 1 Densities of geometrically necessary dislocations and stress heterogeneity. a** Densities of geometrically necessary dislocations (GNDs) estimated from lattice rotations measured by HR-EBSD. Grains with apparent GND densities uniformly >$10^{15}$ m$^{-2}$ have orientations that result in high noise floors, obscuring the intragranular GND structures[40]. **b** Maps of $\sigma_{12}$ normalised by subtracting the mean value within each grain. Samples MN1 and San382t are single crystals, whereas all other samples are aggregates. Annotations indicate the sample number and final deformation temperature. The compression direction was vertical in the maps of the deformed samples. White areas (<47% of map areas) were not indexed during EBSD acquisition or failed the cross-correlation quality criteria (Methods). Black lines mark grain boundaries with misorientation angles >10°.

**Table 1 Sample details.**

| Sample | Grain size[a] (μm) | Final temperature (°C) | Final confining pressure (GPa) | Final differential stress (GPa) | Final strain rate (s⁻¹) | Finite plastic strain (%) | Notes |
|---|---|---|---|---|---|---|---|
| MN1 | N/A | N/A | N/A | N/A | N/A | 0 | Undeformed single crystal |
| PI-1523s | 6.5 | 1200 | 0.3 | N/A | N/A | 0 | Aggregate isostatically hot pressed in a Paterson apparatus |
| San382t | ~700 | 25 | 6.9 | −3.8 | $2.7 \times 10^{-5}$ | 7.2 | Single crystal deformed by cyclic loading in a D-DIA apparatus |
| San382b | 3.0 | 25 | 6.9 | −3.8 | $2.7 \times 10^{-5}$ | 5.3 | Aggregate deformed by cyclic loading in a D-DIA apparatus |
| San372b | 4.6 | 25 | 9.3 | 4.3 | $1.2 \times 10^{-5}$ | 2.1 | Aggregate deformed by unidirectional loading in a D-DIA apparatus |
| PI-1488 | 9.6 | 1150 | 0.3 | 0.258 | $0.9 \times 10^{-5}$ | 5 | Aggregate deformed in a Paterson apparatus |
| PI-1523 | 5.4 | 1250 | 0.3 | 0.257 | $8.9 \times 10^{-5}$ | 17 | Aggregate deformed in a Paterson apparatus |
| PI-1519 | 4.6 | 1250 | 0.3 | 0.204 | $9.7 \times 10^{-5}$ | 19 | Aggregate deformed in a Paterson apparatus |

[a]Grain sizes were measured using the line intercept method on EBSD maps and a constant stereological scaling factor of 1.5[23,31].

**Probability distributions of residual-stress heterogeneity.** Figure 2 presents probability distributions of normalised $\sigma_{12}$ in each sample. All the deformed samples exhibit markedly broader distributions than the corresponding undeformed control samples. The stress distributions are broader in the deformed aggregates than in the deformed single crystal but all deformed samples contain distributions that extend beyond ±1 GPa. The maximum absolute value of normalised $\sigma_{12}$ in the deformed single crystal is 3.3 GPa, whereas those of the deformed aggregates are in the range 7.4–13.5 GPa. These maximum values are consistent with the yield stress of olivine at length scales on the order of 1 μm and room temperature[32]. The probability distributions of normalised $\sigma_{12}$ in the aggregates deformed at low temperatures are similar to, or broader than, those of aggregates deformed at high temperatures. The peak of the probability distribution in the deformed single crystal is not centred on zero due to the slight skew of the distribution. This skew would potentially, but not necessarily, be eliminated if the map covered a larger area. Nonetheless, Fig. 3 of Wallis et al.[37] demonstrates that the map in Fig. 1 is broadly representative of the microstructure of the whole sample. The peaks of the probability distributions in all other samples are centred on zero.

Figure 3 presents a test of the robustness of the probability distributions in Fig. 2 against variations in the size of the stress datasets by plotting peak heights (Fig. 3a) and full widths at half maximum (Fig. 3b) as functions of the number of points included in the dataset. Peak heights vary little when the number of data points exceeds ~$5 \times 10^3$ (Fig. 3a) and full widths at half maximum vary little when the number of data points exceeds ~$10^4$ (Fig. 3b). As the full datasets contain between ~$2 \times 10^4$ and $4.8 \times 10^5$ measurements (Table 2, Methods), the probability distributions plotted in Fig. 2 are likely to vary little with further increases in data size.

Figure 4 characterises the probability distributions of normalised $\sigma_{12}$ in each sample. Fig. 4a presents a 'normal probability plot', in which the cumulative probability distribution is plotted with the probability axis scaled such that a normal distribution falls on a straight line (Methods). This presentation demonstrates that the central, low-stress portion of each distribution falls on a straight line corresponding to a normal distribution. However, each distribution also exhibits a 'tail' that deviates from the straight line to higher stress magnitudes. In the deformed single crystal and aggregates, the magnitude of this deviation is greater than in the corresponding control samples. These tails correspond to stress heterogeneities typically greater than 1 GPa in magnitude.

The plot of restricted second moments ($\nu_2$) of the stress distributions in Fig. 4b characterises the form of the tails of the probability distributions (Methods). All samples exhibit significant portions of their probability distribution that fall on straight lines in Fig. 4b, indicating that $P(\sigma_{12}) \propto |\sigma_{12}|^{-3}$, as expected of stress fields generated by dislocations[44–47] (Methods). Several of the curves depart from straight lines at the highest stresses due to averaging of the elastic strains over the finite volume illuminated by the electron beam[45]. In this plot, the curves corresponding to the deformed single crystal and aggregates exhibit steeper gradients than those of their corresponding control samples, consistent with the presence of greater total dislocation densities in the deformed samples (Methods). The aggregates deformed at 25 °C exhibit distributions comparable to those deformed at 1150–1250 °C in both Fig. 4a, b. One of the aggregates deformed at 25 °C, sample San372b, exhibits the greatest stress heterogeneity.

Figure 5 provides a deeper analysis of the restricted second moments of the stress distributions in the aggregates. Here, the stress data are divided into subsets based on the corresponding

**Table 2 Dataset details.**

| Sample | Number of map points | Step size (μm) | Number of pixels in diffraction patterns | Pixels indexed during EBSD (% of full map) | Pixels in stress datasets passing the HR-EBSD quality criteria (% of full map) |
|---|---|---|---|---|---|
| MN1 | 400 × 500 | 0.5 | 1344 × 1024 | 100 | 100 |
| PI-1523s | 450 × 650 | 0.2 | 1344 × 1024 | 97 | 96 |
| San382t | 155 × 295 | 0.15 | 1344 × 1024 | 100 | 100 |
| San382b | 320 × 200 | 0.15 | 1344 × 1024 | 97 | 91 |
| San372b | 300 × 450 | 0.2 | 640 × 480 | 92 | 69 |
| PI-1488 | 575 × 850 | 0.2 | 640 × 480 | 96 | 89 |
| PI-1523 | 116 × 172 | 0.2 | 1344 × 1024 | 96 | 96 |
| PI-1519 | 600 × 800 | 0.2 | 640 × 480 | 95 | 93 |

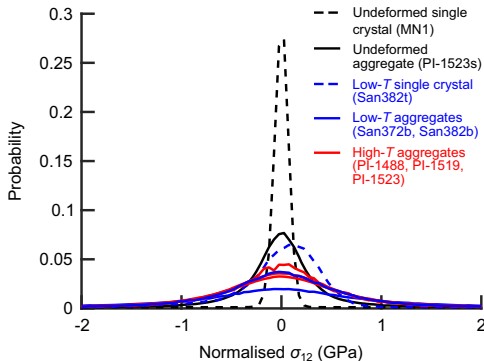

**Fig. 2 Probability distributions of stress heterogeneity.** Probabilities of stress heterogeneity, represented as normalised $\sigma_{12}$, were calculated using bin widths of 50 MPa.

GND density at each pixel and the restricted second moments are computed for each subset. Similar to the full datasets in Fig. 3b, each subset in Fig. 4 exhibits a linear portion at high stress magnitudes. Furthermore, Fig. 4 reveals two additional characteristics that are consistent across the six samples. First, the gradients of the linear portions of the curves increase systematically with increasing GND density. Second, the subsets with higher GND densities typically extend to higher stress magnitudes than those with lower GND densities.

**Autocorrelation of residual-stress fields**. Figure 6 presents the autocorrelation functions of normalised $\sigma_{12}$ in the full map area of each sample (Methods). Sample MN1 exhibits a relatively flat autocorrelation function with negligible long-range correlation in its stress field and a maximum off-peak value of 0.1. All other samples exhibit peaks in their autocorrelation functions. The widths of the portions of these peaks with values >0.1 are within the range 1.4–2.2 μm. Sample San382t, a single crystal, exhibits pronounced anisotropy in its autocorrelation function, corresponding to the banded structure in its stress field. All of the aggregates exhibit approximately isotropic autocorrelation functions.

## Discussion

**Dislocation-induced stress heterogeneity**. The results of this study reveal several first-order characteristics of the residual stress fields of deformed olivine beyond those observed in previous work. Compared to previously reported stress heterogeneities on the order of 1 GPa in samples deformed at room temperature[37] (including San382t and San382b), single crystals of olivine deformed at 1000 and 1200 °C exhibit stress heterogeneities with lesser magnitudes, on the order of a few hundred megapascals[38]. A surprising result of the present study is that, in contrast to

single crystals, the aggregates of olivine deformed at 1150–1250 °C exhibit stress heterogeneity with magnitudes (e.g., full widths at half maximum, Fig. 3) again frequently on the order of 1 GPa, closely comparable to the aggregates deformed at room temperature (Figs. 1, 2, 4, and 5).

One hypothesis for the cause of increased residual stresses in aggregates, relative to single crystals, deformed at high temperature is that they are imparted by anisotropic thermal contraction. However, the sample isostatically hot pressed at 1200 °C and cooled without high-temperature deformation exhibits less stress heterogeneity than the samples deformed at high temperatures (Figs. 1, 2, and 4). Similarly, stresses imparted by the 300 MPa of decompression should be small relative to stresses observed in the aggregates deformed at room temperature but decompressed by 6.9–9.3 GPa[31]. Moreover, detailed analysis of the probability distributions of stress heterogeneity indicates that the high-magnitude stresses are instead imparted by dislocations. The stress distributions of samples deformed at both low and high temperatures exhibit high-stress tails that deviate from normal distributions (Figs. 4a and 5) and are typical of materials deformed by dislocation-mediated mechanisms, even at low temperatures[44–47,49]. Importantly, the analysis of restricted second moments of the probability distributions indicates that the tails of the distributions follow $P(\sigma_{12}) \propto |\sigma_{12}|^{-3}$ in all the deformed samples, as expected of stress fields generated by dislocations[44–47] (Methods). The similarities among the probability distributions of residual stress in samples deformed at low and high temperatures, and the particular form of those distributions, provide strong evidence that the high magnitudes of residual stresses recorded by the samples deformed at high temperatures are also predominantly imparted by the dislocation content. The alternative causes of residual stress, that grains interact by mutual exertion of forces on one another due to heterogeneous deformation, thermal contraction and/or decompression, cannot generate this combination of characteristics.

The interpretation that dislocations are the dominant cause of stress heterogeneity in the aggregates deformed at high temperatures is consistent with the previous analysis of single crystals of olivine deformed at similar temperatures[38]. During experiments on single crystals, there are no neighbouring grains to generate stresses from grain interactions, and therefore any stress heterogeneity must result from the dislocation content. The single crystals were deformed at similar final macroscopic differential stresses (218 and 388 MPa) to the aggregates deformed at high temperatures in this study (204–258 MPa, Table 1), and therefore the total dislocation densities of the two sets of samples should be similar[50,51]. However, in contrast, the GND densities observed here (commonly >$10^{14}$ m$^{-2}$, Fig. 1) within the aggregates deformed at high temperatures are orders of magnitude higher than those in the single crystals deformed at high temperatures (on the order of $10^{12}$ m$^{-2}$)[38]. The difference in GND densities is consistent with well-developed theory for the

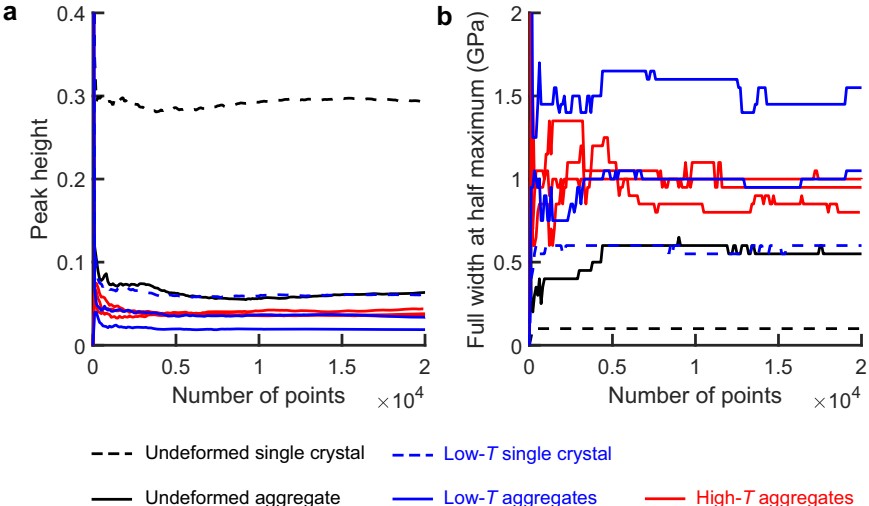

**Fig. 3 Dependence of stress distributions on the size of the dataset. a** Peak heights and **b** full widths at half maximum of the probability distributions of normalised $\sigma_{12}$ as functions of the number of points in the dataset.

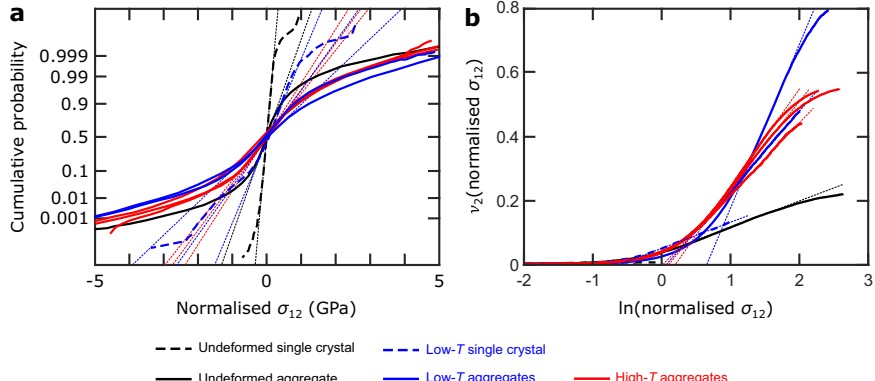

**Fig. 4 Analysis of the form of the stress distributions. a** Normal probability plot of normalised $\sigma_{12}$. On this plot, the cumulative probability axis is scaled such that normal distributions fall on a straight line (indicated as fine dotted lines for visual guides), therefore, deviations from straight lines indicate departures from normal distributions. Solid and dashed lines are calculated from HR-EBSD data. **b** Restricted second moments ($\nu_2$) of the normalised $\sigma_{12}$ probability distributions. On this plot, straight lines (indicated as fine dotted lines for visual guides) indicate that probability $P(\sigma_{12}) \propto |\sigma_{12}|^{-3}$, as expected of stress fields generated by dislocations[44-47]. Solid and dashed lines are calculated from HR-EBSD data.

grain-size dependence of GND density[52–54], in which strain-compatibility constraints imposed by neighbouring grains cause a greater fraction of the total dislocation density to manifest as GND density in finer grained materials. Unlike statistically stored dislocations, the stress fields of GNDs include a significant component that does not cancel over length scales greater than the average dislocation spacing. Therefore, differences in GND density explain the observed differences in the magnitudes of stress heterogeneity between single crystals and aggregates deformed at both low and high temperatures (Figs. 1, 2, 4, and 5)[38].

The associations between total dislocation density, GND density and normalised stress magnitude are further elucidated by the trends evident in Fig. 5. Subsets of stress data from regions with progressively increasing GND density exhibit steeper gradients in the plots of $\nu_2$ versus $\ln(\sigma_{12})$ (Fig. 5). If the slope of $\nu_2$ versus $\ln(\sigma_{12})$ is a reliable proxy for total dislocation density (Methods), this effect indicates correlation between GND density and total dislocation density. The subsets with higher GND densities also typically extend to higher stress magnitudes (Fig. 5). Overall, these observations imply positive correlations among total dislocation density, GND density and normalised stress magnitude. These

associations are evident in the samples deformed at 25 °C, those deformed at 1150–1250 °C, and the isostatically hot pressed sample (which also contains subregions with elevated GND density, Fig. 1). Notably, these relationships manifest at the scale of heterogeneity within individual samples and may or may not apply to comparisons between samples depending on the specific dislocation configurations.

The interpretation that the residual stress distribution is modified by GND density is consistent with the role of GNDs in generating long-range internal stresses[54,55]. Correlation lengths on the order of 1 μm (Fig. 6), along with the spatial distributions evident in the stress maps (Fig. 1b), demonstrate the presence of long-range internal stresses in all the deformed samples. Similar characteristic length scales of stress heterogeneity occur in the single crystals of olivine deformed at high temperature[38] and around nanoindents placed in single crystals of olivine at room temperature[37,40]. Likewise, dislocation dynamics simulations of deforming olivine generate stress heterogeneity over length scales on the order of 1 μm[56]. Therefore, long-range internal stresses appear to be common in deformed olivine, even among samples deformed over a wide range of temperatures up to at least 1250 °C at typical experimental stresses and strain rates. This is by no

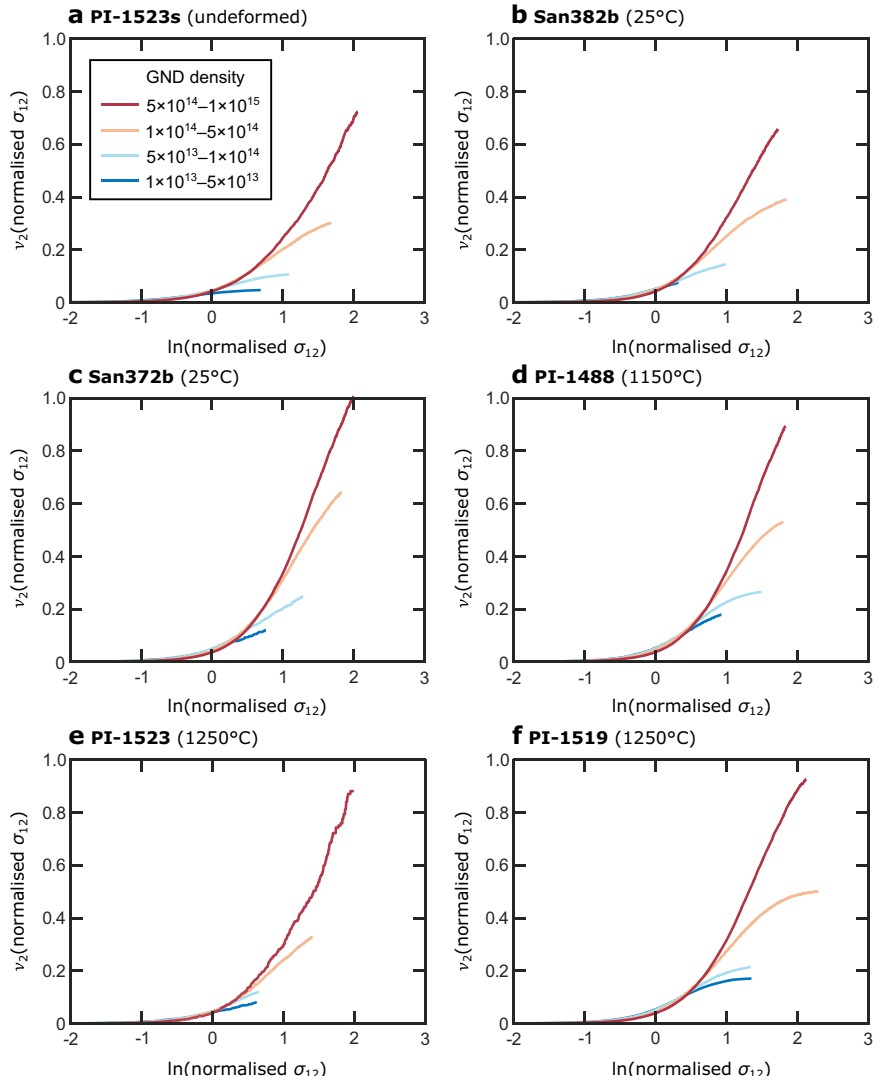

**Fig. 5 Restricted second moments of subsets of the stress distributions based on GND density.** Restricted second moments ($v_2$) of the normalised $\sigma_{12}$ probability distributions in each aggregate (**a**–**f**). The $\sigma_{12}$ data from each sample are divided into subsets based on their corresponding GND densities, indicated in **a**, in the HR-EBSD maps.

means a foregone conclusion for at least three reasons. First, if the plastic-strain fields were typically near homogeneous, most dislocations would be statistically-stored dislocations and their long-range stress fields would largely cancel. Second, strain hardening can occur by short-range dislocation interactions (e.g., formation of junctions) that do not require long-range internal stress[33]. Third, recovery mechanisms, such as subgrain-boundary formation and (sub)grain-boundary migration, can potentially reduce GND densities and long-range internal stresses within subgrain interiors. Future work should address the scaling of long-range internal stresses at lower strain rates and even higher temperatures, under which conditions such recovery mechanisms are more effective at reducing GND density in subgrain interiors.

**Implications for models of transient creep.** The presence of long-range internal stress has been explicitly linked to the transient mechanical behaviour of samples deformed at low temperatures[31]. Both single crystals and aggregates of olivine deformed at room temperature exhibit a Bauschinger effect and, in particular, the relative magnitudes of strain hardening and yield stresses during cyclic loading demonstrate that hardening is dominantly kinematic[31]. Kinematic hardening results from the

action of back stress, generated by long-range elastic interactions among dislocations, that counteracts the applied stress[31]. Whilst the back stress is parameterised as a single scalar value in mathematical formulations of strain hardening[31], its physical manifestation in the material is in the form of long-range stress heterogeneity[57]. Therefore, although it is difficult to calculate the effective back stress from observed stress heterogeneity and vice versa, the mechanical data[31] and microstructural observations (Figs. 1, 2, and 4–6)[37,38], are consistent in indicating the role of long-range dislocation interactions in generating kinematic strain hardening.

Long-range internal stresses have been interpreted to play a key role in establishing steady-state creep by dislocation-accommodated grain boundary sliding, which was inferred by Hansen et al.[23] to be the dominant deformation mechanism in the aggregates deformed at high temperatures that we investigated. During dislocation-accommodated grain boundary sliding, strain compatibility among grains is maintained by dislocation motion in grain interiors, with the rate of dislocation motion limiting the macroscopic strain rate[23,58]. The observed mechanical behaviour (in terms of sensitivity to applied stress and grain size)[23] is consistent with the predictions of microphysical models

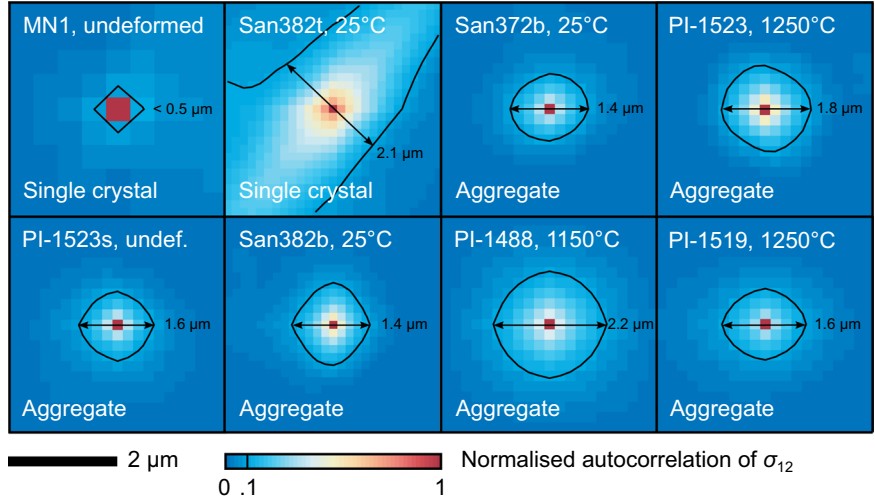

**Fig. 6 Autocorrelation functions of stress heterogeneity.** Autocorrelation functions of stress heterogeneity, normalised $\sigma_{12}$, are normalised to unit peak height. The central $4 \times 4$ μm of the autocorrelation function is plotted for each sample. Black contours correspond to values of 0.1, which is the maximum off-peak value present in the autocorrelation function from the undeformed single crystal, sample MN1. Marked dimensions indicate the width of the region within the 0.1 contour. Annotations indicate the sample number, final deformation temperature and whether the sample is a single crystal or an aggregate.

in which glide is impeded by generation of back stresses among dislocations as they pile up against subgrain boundaries[58]. Our new observation of long-range internal stresses generated by the dislocation content in these samples is consistent with this model (Figs. 1, 2, and 4–6). Long-range dislocation interactions also play a key role in many models of dislocation creep of single crystals at high temperatures[59–61]. The mounting evidence that back stresses among dislocations control strain hardening at low temperatures and steady-state creep at high temperatures raises the question of whether they also make a significant contribution to transient creep at high temperatures.

Current models for transient creep are commonly formulated based on stress transfer between weak and strong slip systems to maintain strain compatibility in an aggregate of mechanically anisotropic grains[12,29]. The initial increments of strain, upon application of macroscopic differential stress, are likely to be accommodated by grains well oriented for glide on the weakest slip system. Due to compatibility constraints, continued deformation on the weakest slip system progressively transfers load to the stronger slip systems until they are sufficiently active to result in steady-state creep. This model was originally formulated for ice[12] and has since been applied to olivine[29], and has inspired recent formulations that are incorporated into large-scale models of postseismic deformation[1,2]. Those formulations describe the viscosity evolution based on combinations of the steady-state flow laws for single crystals of different orientations[1,2]. However, it is important to recognise that, along with aggregates, single crystals deformed at high temperatures also exhibit strain-hardening transient creep (except when limited by the rate of dislocation multiplication, which results in strain softening)[9,33–35]. Therefore, transient creep of an aggregate includes some contribution from intragranular processes that is not captured in the steady-state flow laws typically incorporated in descriptions of postseismic transient creep. At the small strains involved in postseismic deformation (often on the order of microstrain), the transients caused by intragranular processes (i.e., hardening of each slip system) are likely important throughout the postseismic interval but certainly dominate the very earliest deformation that must proceed for the transfer of stresses among grains to ensue. However, previous attempts to model transient creep of single crystals of olivine have been largely limited to fitting phenomenological models

due to a lack of constraints on the underlying microphysical processes[9].

Our results highlight the relevant microphysics associated with transient creep of olivine aggregates. We suggest that the storage and release of back stresses among dislocations provides a conceptual basis for the development of a new generation of models for the contribution of intragranular processes to transient creep of olivine aggregates at high temperatures. This hypothesis is based on the role of back stresses among dislocations in generating strain hardening at low temperatures[31,37], the role of back stresses in steady-state deformation at high temperatures[23,58], and the observation of stress heterogeneity with similar magnitudes and length scales generated by the dislocation content in both sets of samples (Figs. 1, 2, and 4–6). Although there can be differences in the types, densities and/or distributions of dislocations generated at low and high temperatures[30,33,34,36–38,50], the stress fields of individual dislocations have negligible temperature dependence (only that of the shear modulus). Importantly, the new results (Figs. 1, 2, and 4–6) demonstrate that there can be close similarity in the net stress fields of the overall populations of dislocations generated in each temperature regime. Therefore, there is similar potential for long-range interactions among the 'free' dislocations within cell or subgrain interiors.

Although postseismic stress changes affect rocks in both low- and high-temperature regimes in the lithosphere and asthenosphere, our analyses indicate that the same microphysics controls transient behaviour in both cases. This commonality offers potential for a model of transient deformation that is applicable across a variety of settings and conditions. Hansen et al.[31] calibrated a set of constitutive equations for transient low-temperature plasticity based on storage and release of back stress among dislocations. This formulation therefore provides a basis for modelling transient creep at high temperatures if the impact of additional recovery mechanisms, such as dislocation climb, can be taken into account. The form of a new model for transient creep should exploit these new constraints on the underlying processes by incorporating a system of equations derived from the microphysics of dislocation glide, recovery, and/or the evolution of back stress. Specifically, our microstructural observations imply that a quantitative model for high-temperature transient creep should include a back-stress term

that is subtracted from the applied stress so that dislocation glide proceeds under the action of an effective stress[31]. With such a formulation, analogous to that employed for low-temperature plasticity by Hansen et al.[31], changes in applied stress can result in negative effective stresses and therefore generate reverse flow. This viscoelastic behaviour is an important component of recent geodetic analyses of postseismic deformation[1–4,15] and our microstructural observations suggest that it results, at least in part, from back stress generated by long-range dislocation interactions. A model for transient creep based on these intragranular processes could be compared to experimental data from aggregates to test whether additional processes, such as grain interactions[29], contribute significant additional effects. Ultimately, the development of a model for transient creep that can be explicitly related to specific key microphysical processes will provide more robust estimates of the evolution of mantle viscosity over the earthquake cycle. Moreover, by identifying the characteristics of stresses heterogeneity (i.e., the form of the probability distributions, typical length-scales and spatial distributions) in experimental samples deformed at high temperatures, we provide a new set of criteria against which to compare natural rocks to test the relevance of associated models of transient creep to the upper mantle.

Aggregates of olivine deformed at temperatures of 1150–1250 °C exhibit intragranular stress heterogeneity after quenching to room temperature that is remarkably similar in both magnitude (Figs. 1b, 2, 4, and 5) and correlation length (Fig. 6) to that exhibited by aggregates deformed at room temperature. The form of the probability distributions of internal stress (Figs. 2, 4, and 5), along with comparable distributions (albeit of lesser magnitude) in single crystals (Figs. 1, 2, and 4)[38], indicate that the high-magnitude stresses are generated primarily by the dislocation content. The difference in magnitudes of stress heterogeneity between single crystals and aggregates is consistent with the role of GNDs, which occur in higher densities in the aggregates (Fig. 1a), in generating long-range internal stress[53–55]. These observations contribute to a growing body of evidence[31,37,38] suggesting that the storage and release of back stresses among dislocations may be a significant cause of transient creep that is currently not incorporated in models of postseismic deformation. The formulation of new models for these intragranular processes will refine the predictions of large-scale geodynamic simulations.

## Methods

**Deformation experiments**. We utilise samples deformed in two sets of experiments described in detail by Hansen et al.[23,31]. Key information on the deformation conditions is summarised in Table 1. As the deformation conditions typically varied during the experiments (e.g., to obtain flow-law parameters), we report the final conditions in Table 1.

Samples San382t and San382b were deformed at room temperature whilst paired together in a single assembly in a deformation-DIA apparatus[31]. High confining pressures inhibited fracture, whilst differential stresses of several gigapascals induced deformation by low-temperature plasticity. This sample was subjected to shortening followed by extension and exhibited strain hardening of several gigapascals in both cases[31]. The microstructures, including stress heterogeneity, lattice misorientation and distributions of dislocations, of these samples have been characterised in detail by Wallis et al.[37]. The samples contain straight dislocations, often arranged in slip bands, that are spatially associated with stress heterogeneity. Sample San372 was deformed in a similar manner but subjected only to shortening, rather than cyclic loading[31]. We also note that these samples do not exhibit any reverse plastic strain during the final unloading[31], which suggests that the dislocation microstructures prior to unloading are preserved in the final sample material.

Samples PI-1488, PI-1523 and PI-1519 were deformed at final temperatures of 1150 °C and 1250 °C and confining pressures of 300 MPa in a gas-medium Paterson apparatus[23]. Based on the mechanical data and microstructures, Hansen et al.[23] inferred that these samples deformed by dislocation-accommodated grain boundary sliding. Whilst this deformation mechanism involves sliding on grain boundaries, the majority of strain is accommodated by dislocation motion in grain interiors. The samples contain abundant subgrain boundaries and more diffuse

lattice misorientation[23]. At the end of these experiments, the samples were quenched to below 800 °C in less than 300 s while maintaining the final load. This procedure has been demonstrated to preserve the steady-state dislocation microstructure during quenching[50].

We also analyse two samples that act as undeformed controls. The first is a single crystal of San Carlos olivine. The mapped area lacks subgrain boundaries and the crystal contains a low density, $<10^{10}$ m$^{-2}$, of free dislocations[48]. The second is a portion of the starting material, sample PI-1523s, used for experiment PI-1523. The specimen was extracted after sample fabrication by isostatic hot pressing of powdered San Carlos olivine at 1200 °C but prior to the deformation experiment.

**Sample preparation and data acquisition**. Deformed samples were sectioned parallel to the loading axis and prepared for electron backscatter diffraction (EBSD) analysis. The cut surfaces were ground flat and polished with successively finer diamond down to a grit size of 0.25 μm. The surfaces were finished with either 0.05 μm diamond or 0.03 μm colloidal silica. Samples San382t and San382b were coated with 0.5 nm Pt/Pd. The remaining samples were left uncoated.

EBSD data were acquired in two field emission gun scanning electron microscopes (SEMs) equipped with Oxford Instruments AZtec acquisition software and NordlysNano detectors. Samples San382t and San382b were analysed under high vacuum in a Philips XL-30 SEM at Utrecht University. The remaining samples were analysed at low vacuum (50 Pa of water vapour) in an FEI Quanta 650 SEM at the University of Oxford. Reference-frame conventions were validated following the approach of Britton et al.[62] and the microscopes were calibrated for HR-EBSD following the approach of Wilkinson et al.[41]. Table 2 presents the details of each dataset. The footprint of the source region of the diffraction patterns on the specimen surface is estimated to be <100 nm across. Step sizes were in the range 0.15–0.5 μm, which are significantly smaller than the grain sizes of the samples (3–700 μm, Table 1) and are therefore suitable for resolving intragranular GND densities and stress heterogeneity. Initial indexing rates were in the range 92–100% (Table 2). Nonindexed pixels were commonly located on grain boundaries. Nonindexed pixels with at least seven indexed neighbouring pixels within the same grain were filled with the average orientation of the neighbouring pixels.

**High-angular resolution electron backscatter diffraction**. We mapped densities of GNDs and intragranular stress heterogeneity using HR-EBSD postprocessing of the diffraction patterns. A comprehensive overview of the method and its application to geological materials is provided in our recent review paper[40]. Here we provide a summary of key points relevant to the interpretation of the present results. HR-EBSD measures lattice rotation and elastic strain heterogeneity by using cross-correlation to track small shifts in features within EBSD patterns[40–43]. Following acquisition of the EBSD data, one diffraction pattern was chosen from each grain, typically from a location with high pattern quality, to serve as a reference pattern for that grain. One hundred regions of interest, 256 × 256 pixels across, were extracted from each diffraction pattern and a cross-correlation procedure, performed in Fourier space, was used to determine the translational shifts that map the regions of interest in the reference diffraction pattern to their corresponding positions in the other diffraction patterns within the same grain. The displacement-gradient tensor was determined for each diffraction pattern by fitting the shifts measured by cross-correlation. The symmetric and antisymmetric parts of the displacement-gradient tensor describe elastic strains and lattice rotations, respectively. As such, the elastic strains (with associated stresses) and lattice rotations (with associated GND densities) are mathematically independent and do not necessarily occur simultaneously in the material or HR-EBSD data. To allow measurement of the small distortions due to elastic strain in the presence of much larger distortions due to lattice rotations, we employed a two-pass cross-correlation procedure[42]. This procedure uses rotations measured during the first pass to rotate each pattern into coincidence with the reference pattern before measuring elastic strains and small corrections to the rotations during the second pass. Residual stresses were calculated from the elastic strains using Hooke's law and the anisotropic stiffness tensor for olivine at room temperature and pressure[63]. After the cross-correlation procedure, we filtered out results from the stress datasets for which the normalised peak in the cross-correlation function was <0.3 and those with a mean angular error in the deformation gradient tensor >0.004[43]. The final stress datasets include at least 69% of the pixels in each map (Table 2). For datasets of GND density, we filtered out both those pixels that failed the quality criteria and their neighbouring pixels to remove GND densities that were calculated from potentially spurious orientation gradients involving the poor-quality pixels. Most of the pixels that failed the quality criteria were located in intragranular regions with large lattice rotations of at least several degrees relative to the lattice orientation at the reference point.

GND densities were estimated from the spatial gradients in the lattice rotations using the approach of Wallis et al.[48]. This procedure finds the densities of six end-member dislocation types that best fit the measured lattice curvature[48]. Grains with lattice orientations that are unfavourable for the observation of lattice curvature generated by one or more of these dislocation types can exhibit high noise levels that manifest as GND densities $>10^{15}$ m$^{-2}$ over much of the grain (e.g., rare grains in Fig. 1a)[40,48]. However, for the majority of grain orientations, HR-EBSD can resolve GND densities down to a noise level on the order of $10^{13}$ m$^{-2}$ at the step sizes of 0.15–0.5 μm employed in this study (Table 2)[48]. This spatial resolution allows local intragranular regions of elevated GND density to be resolved, whereas

regions of low GND density may be obscured by noise. Furthermore, the measurements of lattice curvature may reveal the presence of dislocations that are not illuminated by other techniques, such as dislocations that did not thread to the specimen surface during oxidation decoration (e.g., Fig. 6 of Wallis et al.[48]). These combined effects can result in apparent GND densities that are higher, particularly in local regions, than average dislocation densities measured by other techniques[48]. Nonetheless, HR-EBSD can resolve a significantly greater fraction of the lattice curvature than conventional EBSD and correspondingly can resolve a greater fraction of the GND content[40,48].

Elastic strain and hence stress are measured relative to the strain state at a reference point chosen within each grain. In deformed materials, it is likely that no part of the material is free from elastic strain, so the results provide maps of relative intragranular strain/stress heterogeneity. In other words, the strain/stress measured at each point is the strain/stress difference from the (unknown) strain/stress state at the corresponding reference point. To provide an intuitive metric that is independent of the choice of reference pattern, we normalise the stresses by subtracting the mean value of each component within each grain[38,49,64]. Therefore, the resulting maps document stress heterogeneity relative to the mean stress state of each grain, which we refer to simply as normalised stresses, **σ**, herein. We have previously used the map of sample PI-1523 to document the effect of this normalisation procedure, which is illustrated in Fig. 6 of our recent review paper[40]. As a different reference point is chosen within each grain, the absolute magnitudes of the stresses may differ by an unknown amount between grains. Nonetheless, the measurements are still sensitive to stresses arising from grain interactions in an aggregate of grains (i.e., the forces of grains indenting their neighbours), which will manifest as intragranular stress concentrations, as well as the intragranular stress fields of dislocations.

HR-EBSD data were acquired on two-dimensional free surfaces of the specimens, which imposes constraints on the measurements of GND density and stress heterogeneity that can be obtained. Gradients in lattice orientation in the direction normal to the specimen surface cannot be observed and therefore dislocations that generate orientation gradients only in this direction cannot be detected[40,48]. Therefore, the measured GND densities represent lower bounds on the total GND content. The presence of a free surface also relaxes the normal stress acting on that surface, while the other stress components are modified to a lesser extent due to the Poisson effect and changes in the tractions on the surface[40,65,66]. Here, we focus on the $\sigma_{12}$ component as this component is least modified by sectioning the samples and generally most closely pertains to the glide forces on dislocations during deformation.

**Analysis of stress heterogeneity.** To test the robustness of the probability distributions of normalised $\sigma_{12}$ against variations in the size of the dataset, we analyse subsets of the full datasets that include varying fractions of the full data. We plot the peak heights and full widths at half maximum of the probability distributions of normalised $\sigma_{12}$ as functions of the number of data included. These plots allow identification of an approximate threshold dataset size above which the characteristics of the probability distributions become approximately invariant with further increases in size.

To characterise the probability distributions of the normalised stresses we use normal probability plots and calculate the restricted second moments of the probability distributions. A normal probability plot displays the cumulative probability with the probability axis scaled such that a normal distribution falls on a straight line. Departures from a straight line indicate departures from a normal distribution and are commonly observed at high stress magnitudes in deformed materials. Work on Cu, InAlN and steel has demonstrated that the stress fields of dislocations typically cause these high-stress 'tails' in the probability distributions[46,49].

Analysis of the restricted second moment of the probability distribution of normalised $\sigma_{12}$ offers a powerful means to test the causal relationship between the dislocation content and high-stress tails in the probability distributions. Although the stress field of a population of dislocations will be the sum of the stress fields of the individual dislocations, which decay linearly with distance from the dislocation core, the analysis is simplified by assuming that an individual dislocation dominates the stress field within a patch in its immediate vicinity. The probability distribution of stress within these patches has a predictable form. A remarkable property of a population of straight parallel dislocations is that, regardless of their spatial configuration, the probability distribution of its stress field tends to $P(\sigma) \propto |\sigma_{12}|^{-3}$ at high stresses, following

$$P(\sigma) \rightarrow C\rho |\ \sigma\ |^{-3} \qquad (1)$$

where $C$ is a constant that depends on the material, type of dislocation and considered stress component, and $\rho$ is the total dislocation density (i.e., including both geometrically necessary dislocations and statistically stored dislocations)[44–47]. To test whether the measured stress fields exhibit this characteristic form we compute the restricted second moment, $v_2$, which is a metric that characterises the shape of a probability distribution, $P(\sigma)$, based on the integral over restricted ranges in stress[46], calculated as

$$v_2(\sigma) = \int_{-\sigma}^{+\sigma} P(\sigma)\sigma^2 \mathrm{d}\sigma \qquad (2)$$

As an important consequence, a plot of $v_2$ versus $\ln(\sigma_{12})$ becomes a straight line at high stresses if the stress field exhibits the form $P(\sigma_{12}) \propto |\sigma_{12}|^{-3}$ expected of a population of dislocations, and the gradient of that line is proportional to the dislocation density[45,46]. Whilst $C$ can be determined for simple populations of dislocations (e.g., one type of edge dislocation)[44–47], no theory has been developed for more complex populations of dislocations. Therefore, we do not attempt to quantify $\rho$ in our samples, which contain multiple types of dislocations on different slip systems[23,37], oriented differently with respect to the sample reference frame to which the stress is referred. However, we do use the relative magnitudes of these gradients to imply an approximate ranking in the total dislocation densities among samples. In particular, we utilise this effect to explore the relationships between total dislocation density (from the gradient of $v_2$ versus $\ln(\sigma_{12})$), GND density (calculated from the orientation gradients) and normalised stress magnitude. To do so, we subset the stress data based on the corresponding GND density at each pixel and plot $v_2$ versus $\ln(\sigma_{12})$ for each subset. These plots provide a test for systematic variations in gradient (i.e., total dislocation density) with subset (i.e., GND density). We perform this analysis for each aggregate but not for the single crystal, as the former exhibit a sufficient range of GND densities.

To characterise the length scales of stress heterogeneity, we employ autocorrelation functions of normalised $\sigma_{12}$. We compute the autocorrelation functions from the full map area of each sample. The autocorrelation function is the integral of the product of two superposed copies of the stress map that are progressively displaced from one another. Spatial correlation (e.g., stresses of the same sign likely being adjacent) generates a peak in the autocorrelation function. The width of the peak indicates the characteristic correlation length of the stress field. To quantify this value we measure the width of the portion of the peak that exceeds the maximum off-peak value (0.1) observed in the autocorrelation function of the undeformed single crystal, sample MN1. The spatial resolutions of the autocorrelation functions are equivalent to the step sizes used to acquire the EBSD data and are therefore in the range 0.15–0.5 μm (Table 2).

## Data availability
The data that support the findings of this study are available from the corresponding author upon reasonable request.

## Code availability
HR-EBSD processing was performed using a restricted developmental code in MATLAB® that performs the analyses detailed by Wilkinson and coworkers[41–43]. The same analyses can be performed using the proprietary software CrossCourt4 distributed by BLG Vantage.

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

## Acknowledgements

This work was funded by Natural Environment Research Council grant NE/M000966/1 to all authors; the Netherlands Organisation for Scientific Research, User Support Pro-gramme Space Research, grant ALWGO.2018.038 to D.W.; and startup funding to D.W. from Utrecht University.

## Author contributions

D.W. and L.N.H. conceived the study and performed deformation experiments. D.W. acquired and processed the microstructural data and drafted the manuscript. D.W., L.H.N., A.J.W. and R.A.L. contributed to data interpretation and revision of the manuscript.

## Competing interests

The authors declare no competing interests.
