## [Peer Review File · Nature Communications]

REVIEWER COMMENTS

Reviewer #1 (Remarks to the Author):

Wallis et al. present a novel new analysis of residual stresses associated with dislocations in deformed olivine aggregates. The analysis of the spatial scales of dislocation interactions provides new context for the development of new models for transient creep in the mantle (the focus of this paper), but is really more general than that. The paper is another step forward in the excellent work conducted by this group – following a parallel trend in the analysis of high-resolution microscopy in material science. The paper would be of interest to a wide audience and I would recommend publication after the authors address the following comments.

1. Comparison of dislocation density to images of oxidized samples: Previous work by the authors shows nice examples of comparisons between dislocations imaged in oxidized olivine samples and the EBSD analysis. In the current paper, I was struck by the very high dislocation densities in the high T samples deformed at relatively low stress (on the order of 100s of MPa), as they appear significantly higher than those reported in the literature (or what is represented by visual inspection of micrographs in published papers) – for example: Farla et al. 2011, 2012; Lee et al., Tectonophysics 2002. Following the examples from Wallis et al. JGR 2017, I think it would be very helpful to present a micrograph of an oxidized sample of the high T experiment (e.g., PI-1523). I understand some GND (for example in subgrains) can be very closely spaced – as stress fields of these dislocations partly “cancel” each other. But the spatial scale where densities on order of $10^{14}/\text{m}^2$ are illustrated does not seem consistent with such structures (for example, relatively “broad patches” with GND $\sim 3 \times 10^{14}/\text{m}^2$ for PI-1523 in Figure 2). I also understand there are some issues with orientation effects on the resolution of the technique. It would be helpful if the authors discussed this issue in more detail in the paper to highlight their confidence in the higher dislocation densities in the high T samples.

2. In presenting the data, I think that showing the GND figures directly next to the stress figures (i.e. individual panels from Figures 1 and 2) would be really helpful (basically combining Figures 1 and 2 into one figure). In Figure 2, I would also like to see the plots before the pixel stress is subtracted by the average stress of the grain.

3a. The analysis of the high stress tails is interesting, but could use a bit more description. The authors follow Wilkinson et al. 2012. I found the description of Figure 3 in that paper extremely helpful for understanding what was presented by the authors in Figure 4. From reading Wilkinson, I understand that the prediction of probability proportional to $1/\sigma^3$ is basically describing the stress field around isolated dislocations. Thus, the analysis of this portion of the curve offers the opportunity to calculate the dislocation density (basically from the slope of the straight part of the curves in figure 4b). As I understand this, the EBSD analysis has high enough resolution to resolve the high stress region around the dislocations (where the stress drops off quickly as $1/\text{distance}$ away from the dislocation). What I am missing here is how to relate this back to the backstress – and eventually the models for transient creep – discussed in general terms by the authors - as the analysis seems to be simply finding the dislocations.

3b. From this context, what do individual grain plots look like on plots of second moment versus $\ln(\sigma)$? The scale of the grains is actually similar to that for the “theoretically-generated” plot shown in Figure 3 of Wilkinson et al. Can you see differences in the tails consistent with the different dislocation densities shown in Figure 1?

4. Autocorrelation: This analysis provides context for the length scale of stress gradients in the sample, which is nice. I am curious about the isotropic nature of these plots. Are they averages over the entire EBSD map? What do the “mini-autocorrelation” plots look like for individual grains in the aggregates? Similar to observations by the authors for single crystals (JGR 2017), it would be interesting to see if anisotropy is evident in the stress fields for individual grains. If not, that is also interesting.

Specific comments

1. Please use scale bars for Figures 1 and 2
2. Line 187: What is the reference for the 6.9 to 9.3 GPa?

Nice work, Greg Hirth

Reviewer #2 (Remarks to the Author):

Review of Manuscript: Dislocation interactions in olivine control postseismic creep of the upper mantle.

Authors: David Wallis, Lars N. Hansen, Angus J. Wilkinson, Ricardo A. Lebensohn

Submitted to Nature Communications

Dear Authors and Editor,

This paper's main statement is that postseismic creep in the upper mantle is likely to be controlled by intragranular, transient dislocation interactions in olivine. The link is made by reporting that high residual stresses in olivine deformed at low temperatures are comparable in magnitude and correlation length to those measured in olivine deformed at high temperatures. In addition, the manuscript emphasises that intragranular dislocation interactions are often overlooked, in favour of the effects of constraints imposed by neighbouring grains.

This study is certainly important because little is known about the effects of transient deformation in the lower crust and upper mantle following an earthquake and during postseismic creep. It is often suggested that transient stress changes resulting from shock events are likely to be accommodated by the plastic response of rocks, namely the motion and interactions of dislocations. However, the Geoscience community is still some way away from understanding the mechanisms that trigger such feedback processes and what are the key controls.

Therefore, I think the manuscript reports an important and much needed study (for the Geoscience and broader community) with a degree of novelty. Nevertheless, I feel the Authors have to an extent overlooked existing relevant literature on strain hardening due to dislocation interactions in their own field (e.g. Mussi et al., 2017, Phil Mag) and in other fields, such as materials science (for example Koslowski et al., 2002, JMPS), favouring citation of their own recent work.

To provide what I hope is a constructive review on the body of the paper, below I report line/figure numbers with my comments.

Introduction

50-52: "the overarching arrangement..." this sentence is not supported by a relevant reference. Give such reference/s. I also found that important statements such as this one are made using specialist terminology, which weakens the clarity of the message to the broader audience, at least in my view.

63: explain why the stress transfer model does not apply. Surely stress transfer between slip systems and the various dislocations structures does apply (see also Mariani et al., 2010 on the deformation of MgO single crystals).

70-72: "This suggestion is supported..." I found this sentence difficult to understand at this point of the introduction. Expand on and clarify what you intend by "indistinguishable strain hardening".

76-81: Sound hypotheses, but at what kind of time-scales are you expecting intragranular dislocation interactions in transient creep to control postseismic deformation? Also, explain why you don't seem to think that neighbouring grain interactions in natural olivine aggregates will have similar controls at the same time as intragranular interactions?

86: I suggest that, for a broader audience, you should explain briefly but clearly what GNDs are and why they are important.

86-95: this last part of the introduction contains unexplained specialist terminology that detract somewhat from the message. Some explanations are given in the method, which helps, but this should be done even further so that all procedures and codes applied are well explained.

Results

Table 1 and 2 in Methods are useful to gather an immediate idea of the datasets available. They should be referred to as soon as relevant in the results.

Figure 1. – A summary such as this: “Fig. 1 is a visual representation of GND density from HAR-EBSD showing results from a variety of deformed and undeformed sample. All low T samples were deformed at high P_c , while all high T samples were deformed at low P_c . All samples, except single crystals, have grain size $<10\mu\text{m}$. Only 2 high T aggregate samples were deformed at $\sim 20\%$ strain and faster strain rates of $\sim 10^{-4}/\text{s}$.” in the caption or in the text would help the reader get to grips with the variety of conditions investigated.

I assume the HAR-EBSD data comes from the area of each individual map in figure 1. One immediate observation is that each map is of a different size and contains considerably different number of grains (for aggregates) and therefore of boundaries too. My concern therefore is that the Authors have not considered how representative of the whole sample microstructure each individual area is.

As additional work, an assessment of the optimal representative area for each sample microstructure should be carried out. This can be done by testing the area threshold by which, above a minimum area size map, the sample characteristics measured remain the same. This investigation is important because it will reduce errors in Figures 3, 4 and possibly 5, will reveal more reliable trends of stress probability distribution between different temperature experiments and will therefore result in more reliable interpretations of sound datasets.

From qualitative observation it is clear in Figure 1 that high P_c , low T deformed samples generally have grains with higher GND density. Has any TEM work been done to document the actual defect content of these grains and compare it with the low P_c , high T deformed samples? TEM would identify dislocations as well as more complex defects, stacking faults, micro-deformation bands, microcracks. It seems important to have structural evidence to back the statement that dislocations interactions maybe the same in the low T and high T samples.

An additional single crystal deformed at high T would have been a good measure for comparison with low T single crystal and high and low T aggregates.

The layout of both Figures 1 and 2 may be improved by adding information on HIPing T for b) and by organising samples by T (e.g. all low T on the left, all high T on the right). Also, in Figure 2 the scale label is unclear, until later on, when one reads in the Methods that residual stress normalisation was done by subtracting the grain mean stress to σ_{12} . I would encourage the Authors to ensure, within space limits, that all figures and concepts are clear and self-sufficient on first read.

137: “the probability distribution of...” add “the normalised residual stress”.

Figure 3: I found it informative (although subject to how representative of each sample each area analysed is, particularly for the deformed aggregates), but I would like to learn which sample is which directly from the figure, without having to try and double guess. You could add this information in the legend. Explain in the caption why the deformed single crystal peak is not centred on 0 GPa like all other peaks.

Something I have been thinking about for a little while is the fact that, at least in my experience, residual stresses measured using HAR-EBSD on a variety of materials deformed at a variety of conditions (e.g. calcite, feldspar, olivine in this study and possibly other materials), seem to be of the order of between 0.5 and 1 GPa or thereabout. I would be interested, out of curiosity, on the Author’s comments on this and the potential reasons, given their experience with the application of HAR-EBSD to deformed samples that display large rotations.

142: Note that Figure 4 shows the cumulative probability distribution of normalised residual stresses, not the normal probability distribution. Correct text and caption. In the same line, distributions are shown for each area analysed, not for each sample. Correct this too.

I find Figure 4 rather confusing for both, how it is represented and how it is explained. It is difficult to pick trends and what they mean/fit with the interpretations. The restricted second moments need to be explained in a way that is accessible to the broader audience. This is true for the Method part at lines 348 onwards too. My advice to the authors here would be that they reflect on the key message they want to convey with these diagrams and redraft diagrams and text to achieve maximum clarity on this.

I have a similar comment for the use of the autocorrelation function in Figure 5: it needs to be explained more clearly (how it is obtained, where is it taken, what does it show). Understanding how the “peak” is defined is important to understanding how the correlation length of the stress field is determined.

Discussion

196-199: it might be that I am missing some key piece of information, but I find it puzzling that the high P_c , low T samples display a high number of grains with high GND density (considerably higher than high T samples) and yet the residual stresses measured in high T and low T samples are comparable. The authors

say that this demonstrates “the high stress magnitudes recorded by the samples deformed at high temperatures are also predominantly imparted by the dislocation content”. Could the Authors explain why such different GND densities should be associated with similarly high residual elastic stresses?

205: check reference cited here as it does not seem to be directly relevant to the statement made as it is concerned with the study of single crystals.

211-213: but then perhaps the same GND density differences don’t seem to explain the similarity in the magnitude of stress heterogeneity between low T and high T aggregates...

225: need to add a reference after “...temperatures.”

237-238: observations may be new for the samples in this particular study but the idea that dislocations interactions generate long range internal stresses is a concept long studied in the materials community and to an extent also in the Geoscience community (e.g. Boioli, Tommasi et al., 2015).

255-258: explain how short lived transients may be important across such an extended time period as suggested in these lines.

261-end: Mussi et al. 2017 and mat sci references such as Koslowski et al., 2002 are missing.

Overall, I found this study a very interesting read (although convoluted at times) and I do think it has the potential to influence the direction of thinking in the wider Geoscience field, however I found the conclusions are based on evidence that should be improved by ensuring analyses are based on representative areas of each sample. TEM work would also be valuable evidence in support of the statement the Authors make that intragranular dislocation interactions result in long range residual stress fields that are similar in olivine deformed at low T and olivine deformed at high T. Finally, as per my comments above, the Authors should clarify a number of important points to strengthen the construction of their interpretations and conclusions. Analyses are based on the code developed by Wilkinson et al., 2006 and Britton et al., 2012, and in parts further developed by the Authors. Similar code and access to samples (or the ability to run experiments) would be needed to be able to reproduce this work.

I hope the Authors will find my review constructive

With best wishes

Dr Elisabetta Mariani

Reviewer #3 (Remarks to the Author):

Dear editor,

You asked me to review “Dislocation interactions in olivine control postseismic creep of the upper mantle” authored by David Wallis, Lars N. Hansen, Angus J. Wilkinson, and Ricardo A. Lebensohn. The authors hypothesize “that the same intragranular processes control transient creep of aggregates of olivine deformed by power-law creep and those deformed by low-temperature plasticity”. To test their hypothesis, they analyze orientation patterns of single crystals and polycrystalline aggregates of olivine, experimentally deformed at different conditions with a focus on the difference in temperature.

Unfortunately, I cannot recommend publication of the manuscript in its current state. In fact, I have doubt about the validity of the applied method. Irrespective of my issues with the method, outlined below, I wonder: “Is testing the hypothesis enough for publication?” In their last sentence of the “introduction”, the authors write:

“If the hypothesis is supported by our results, these observations will provide the basis for a new generation of rheological models of transient creep, rooted in the microphysics of intracrystalline deformation.”

To stop after validation of the hypothesis occurs to me like what one would do when writing a proposal. In a publication, I would like to learn at least how the models would look like and what would change in our perspective of Earth processes once they are at hand? Better even, the authors would formulate a model (based on their observations) and test its predictions against independent observations. To me, however, the manuscript is rather vague about what observations exactly shall be modeled.

The presentational aspects of the manuscript need further attention, as for example clarifying what is presented in the figures. Please find attached a digitally annotated pdf-version of the manuscript addressing

problems with wording etc. The annotations reflect my immediate response when reading the manuscript and I am afraid that –despite some retrospective filtering and modification- my digital annotations/comments are not well organized, to some extent reflecting my difficulties to “get some ground under my feet” due to what I find rather imprecise wording and lack of conciseness. When referring to presentation the comments are meant as examples to be transferred to the entire text. Please accept my apologies for the “sloppy” wording of my digital comments; no offense meant whatsoever.

Kind regards,
Joerg

my concerns about the approach:

The authors try to deduce two quantities from EBDS-patterns

1) densities of (geometrically necessary; nominally sessile) dislocations

2) stress

One may fundamentally question whether it is possible to derive “independently” two different physical quantities from a “single” measurement; I am not saying it is impossible but one had to demonstrate to what extent they are indeed independent or to what extent they exhibit correlations due to the method. To some extent, I am not so sure whether testing the hypothesis actually requires determining both, dislocation density and stress. The main objective, to my understanding, is to demonstrate the similarity of the deformation structures of samples deformed at different temperatures.

My biggest problem relates to the method used to derive strain and stress of individual grains. The methods section is not very clear about the chosen procedure. Actually, looking at the referred earlier publication of the first author provided me with:

“Stored diffraction patterns were reanalyzed using the HR-EBSD approach of Wilkinson et al. (2006) and Britton and Wilkinson (2011, 2012a). A reference pattern with high band contrast was chosen from each map. One hundred regions of interest (ROIs) were selected from each diffraction pattern and cross correlated with the corresponding ROIs in the reference pattern to determine the relative shift of each ROI. A displacement gradient tensor was fitted to the shifts in each pattern, allowing calculation of the lattice rotations and elastic strains relative to the reference pattern, both with a sensitivity of approximately 10^{-4} (Wilkinson et al., 2006).

Residual stress variations were calculated from the elastic strain variations using the elastic moduli of olivine at 1 atm and 273 K (Abramson et al., 1997). In this analysis, the stress component normal to the sectioned specimen surface is assumed to be relaxed to zero. Strains are calculated in the reference frame of the microscope stage, but we present components of the stress tensor in the crystal reference frame to aid interpretation of their impact on dislocation processes.”

Even this description did not dispel my doubts. Microstructural patterns can only be used to derive “spatial variations in strain” and from them “spatial variations in stress”, but neither “strain” nor “stress”. The first step is to derive a displacement-gradient matrix that then has to be separated into its symmetric and anti-symmetric parts. The next step is to calculate the spatial variations of these, from which “spatial variations in strain” can be deduced by “gradient” operations. The final step is to transfer “spatial variations in strain” to “spatial variations in stress” using a rheological model. From the description of the method, it seems that the authors did not do the second step of calculating the spatial variations of the symmetric and anti-symmetric part but treated the symmetric part of the displacement-gradient matrix as “strain” and transferred that to “stress”. The simple example of a twin shows that this “abbreviated” method is wrong. One half of a twin has a huge strain compared to the other and the abbreviated method would infer an according stress though the twinned crystals is free of internal stresses for the most part. For the twin, spatial CHANGES in strain occur only at the twin boundary and the field analysis of changes will correctly reflect the microphysics, from which we know that the remaining misfits of atomic layers at twin boundaries cause only local internal stresses [reflected by the finite interface energy].

If the authors used the correct approach (but did not communicate that well ...), still critical considerations

are missing on:

- 1) What is the effect of the numerical calculations of the spatial fields of interest? Deducing (changes in) strain from EBSD-patterns requires numerical calculation of spatial derivatives from digital data. The correct method actually requires second derivatives. Any numerical calculation of derivatives involves loss of information that results in "smoothing". It is thus critical to examine the limits of interpretation one can apply to the spatial characteristics in the light of the numerical procedure.
- 2) What are the limits of the inherently two-dimensional analysis? The surface of the samples is stress-free at the point of measurement; no information on vertical displacement available.
- 3) What happens to the samples during quenching? What happens to glissile dislocations? How do they affect the structure of sessile dislocations?)
- 4) What is the effect of the convolution of the spatial characteristics of the real misorientation pattern and the resolution (size) of the electron beam? How does "filtering out"/"lack of indexing" for individual spots affect the outcome?

Reviewer's comment	Authors' response
Reviewer 1 I was struck by the very high dislocation densities in the high T samples deformed at relatively low stress (on the order of 100s of MPa), as they appear significantly higher than those reported in the literature (or what is represented by visual inspection of micrographs in published papers) – for example: Farla et al. 2011, 2012; Lee et al., Tectonophysics 2002. Following the examples from Wallis et al. JGR 2017, I think it would be very helpful to present a micrograph of an oxidized sample of the high T experiment (e.g., PI-1523). I understand some GND (for example in subgrains) can be very closely spaced – as stress fields of these dislocations partly “cancel” each other. But the spatial scale where densities on order of $10^{14}/\text{m}^2$ are illustrated does not seem consistent with such structures (for example, relatively “broad patches” with GND $\sim 3 \times 10^{14}/\text{m}^2$ for PI-1523 in Figure 2). I also understand there are some issues with orientation effects on the resolution of the technique. It would be helpful if the authors discussed this issue in more detail in the paper to highlight their confidence in the higher dislocation densities in the high T samples.	There are a variety of technical reasons why GND contents measured by HR-EBSD (particularly within small subregions) may be greater than average dislocation densities measured by other techniques. The two types of data are nonetheless consistent/compatible and we have previously published an example of the comparison that the reviewer suggests. We have added the following paragraph to explain these subtleties... “GND densities were estimated from the spatial gradients in the lattice rotations using the approach of Wallis et al.⁴⁸. This procedure finds the densities of six end-member dislocation types that best fit the measured lattice curvature⁴⁸. Grains with lattice orientations that are unfavourable for the observation of lattice curvature generated by one or more of these dislocation types can exhibit high noise levels that manifest as GND densities $> 10^{15} \text{ m}^{-2}$ over much of the grain (e.g., rare grains in Figure 1a)^{39,48}. However, for the majority of grain orientations, HR-EBSD can resolve GND densities down to a noise level on the order of 10^{13} m^{-2} at the step sizes of 0.15–0.5 μm employed in this study (Table 2)⁴⁸. This spatial resolution allows local intragranular regions of elevated GND density to be resolved, whereas regions of low GND density may be obscured by noise. Furthermore, the measurements of lattice curvature may reveal the presence of dislocations that are not illuminated by other techniques, such as dislocations that did not thread to the specimen surface during oxidation decoration (e.g., Figure 6 of Wallis et al.⁴⁸). These combined effects can result in apparent GND densities that are higher, particularly in local regions, than average dislocation densities measured by other techniques⁴⁸. Nonetheless, HR-EBSD can resolve a significantly greater fraction of the lattice curvature than conventional EBSD and correspondingly can resolve a greater fraction of the GND content^{39,48}.”
I think that showing the GND figures directly next to the stress figures (i.e. individual panels from Figures 1 and 2) would be really helpful (basically combining Figures 1 and 2 into one figure).	Done
In Figure 2, I would also like to see the plots before the pixel stress is subtracted by the average stress of the grain.	We have actually previously presented this data for one of the samples used in the present study, so we have added the text “We have previously used the map of sample PI-1523 to document the effect of this normalisation procedure, which is illustrated in Figure 6 of our recent review paper⁴⁰.” As the extra plots would provide no additional information relevant to the scientific argument of the paper (it should be specifically the normalised stresses that are used in the data analysis), and we have cited four references that can be examined for further examples of the procedure, we have not added the plots to the paper.
The analysis of the high stress tails is interesting, but could use a bit more description. The authors follow Wilkinson et al. 2012. I found the description of Figure 3 in that paper extremely helpful for understanding what was presented by the authors in Figure 4. From reading Wilkinson, I understand that the prediction of probability proportional to $1/\sigma^3$ is basically describing the stress field around isolated dislocations. Thus, the analysis of this portion of the curve offers the opportunity to calculate the dislocation density (basically from the slope of the straight part of the curves in figure 4b). As I understand this, the EBSD analysis has high enough resolution to resolve the high stress region around the dislocations (where the stress drops off quickly as $1/\text{distance}$ away from the dislocation). What I am missing here is how to relate this back to the backstress – and eventually the models for transient creep – discussed in general terms by the authors – as the analysis seems to be simply finding the dislocations.	We have expanded the description of the analysis in the Methods section with the following: “Although the stress field of a population of dislocations will be the sum of the stress fields of the individual dislocations, which decay linearly with distance from the dislocation core, the analysis is simplified by assuming that an individual dislocation dominates the stress field within a patch in its immediate vicinity. The probability distribution of stress within these patches has a predictable form.” We have clarified the link between stress heterogeneity and back stress by adding the following to the Discussion section: “Kinematic hardening results from the action of back stress, generated by long-range elastic interactions among dislocations, that counteracts the applied stress³¹. Whilst the back stress is parameterised as a single scalar value in mathematical formulations of strain hardening³¹, its physical manifestation in the material is in the form of long-range stress heterogeneity. Therefore, although it is difficult to calculate the effective back stress from observed stress heterogeneity and vice versa, the mechanical data³¹ and microstructural observations (Figures 1–4)^{37,38}, are consistent in indicating the role of long-range dislocation interactions in generating kinematic strain hardening.”
From this context, what do individual grain plots look like on plots of second moment versus $\ln(\sigma)$? The scale of the grains is actually similar to that for the “theoretically-generated” plot shown	These are interesting questions. We have addressed them by generating subsets of the stress data based on the corresponding GND densities at each pixel and then plotting the restricted second

in Figure 3 of Wilkinson et al. Can you see differences in the tails consistent with the different dislocation densities shown in Figure 1?	moments of those subsets. We have added a new figure (Figure 4) and corresponding paragraphs in the Results and Discussion sections to answer these questions.
Autocorrelation: This analysis provides context for the length scale of stress gradients in the sample, which is nice. I am curious about the isotropic nature of these plots. Are they averages over the entire EBSD map? What do the “mini-autocorrelation” plots look like for individual grains in the aggregates? Similar to observations by the authors for single crystals (JGR 2017), it would be interesting to see if anisotropy is evident in the stress fields for individual grains. If not, that is also interesting.	The autocorrelation functions are computed from the full map areas and we have added sentences to the Results and Methods section to clarify this. Due to the small grain size of the samples, mini-autocorrelation plots from individual grains would add little additional information compared to visual inspection of the original stress maps. However, we have added a sentence to the Results section to state that there is little anisotropy evident within individual grains in the aggregates.
Please use scale bars for Figures 1 and 2	Done
Line 187: What is the reference for the 6.9 to 9.3 GPa?	Added
Reviewer 2	
Nevertheless, I feel the Authors have to an extent overlooked existing relevant literature on strain hardening due to dislocation interactions in their own field (e.g. Mussi et al., 2017, Phil Mag) and in other fields, such as materials science (for example Koslowski et al., 2002, JMPS), favouring citation of their own recent work. 237-238: observations may be new for the samples in this particular study but the idea that dislocations interactions generate long range internal stresses is a concept long studied in the materials community and to an extent also in the Geoscience community (e.g. Boioli, Tommasi et al., 2015).	We have added references to Mussi et al. (2017) and Boioli et al. (2015) in the earth sciences and Koslowski (2002), Blum and Weckert (1987), Weertman (1968), and Kassner (2015) in the material sciences.
50-52: “the overarching arrangement...” this sentence is not supported by a relevant reference. Give such reference/s. I also found that important statements such as this one are made using specialist terminology, which weakens the clarity of the message to the broader audience, at least in my view.	Reference added
63: explain why the stress transfer model does not apply. Surely stress transfer between slip systems and the various dislocations structures does apply (see also Mariani et al., 2010 on the deformation of MgO single crystals).	We are talking about stress transfer between grains of different orientations (i.e., the model developed for ice by Ashby and Duval, 1985) so we have added “ intergranular stress-transfer model” to clarify.
70-72: “This suggestion is supported...” I found this sentence difficult to understand at this point of the introduction. Expand on and clarify what you intend by “indistinguishable strain hardening”.	We have added the underlined text to clarify “This suggestion is supported by the practically indistinguishable shapes of the stress-strain curves a comparison between strain hardening behaviour of single crystals and that of aggregates of olivine between the yield stress and the flow stress when deformed at room temperature ³¹ . The behaviour of both the single crystals and aggregates can be quantitatively described by a single model based on long-range dislocation interactions ³¹ .”
76-81: Sound hypotheses, but at what kind of time-scales are you expecting intragranular dislocation interactions in transient creep to control postseismic deformation? Also, explain why you don’t seem to think that neighbouring grain interactions in natural olivine aggregates will have similar controls at the same time as intragranular interactions? 255-258: explain how short lived transients may be important across such an extended time period as suggested in these lines.	The transients may occur over small strain intervals but can still take an appropriate amount of time, so there is no conflict here. For example, at a strain rate of 10^{-14} s^{-1} it takes 3 years to generate 1 microstrain, which are all values typical of postseismic deformation. However, these considerations of postseismic deformation in general are somewhat beside the point. All we’re saying is that postseismic deformation involves very small strains (which is well established) and over those strain intervals intragranular processes will contribute to transients. We are explicitly not arguing here that grain interactions do not impact transient creep alongside intragranular processes. In the introduction we state “Whilst the transfer of stress among slip systems in grains of different orientations does potentially contribute to transient creep of rocks...”. We have modified the following sentence in the Discussion to clarify the situation: “At the small strains involved in postseismic deformation (often on the order of microstrain), the transients caused by intragranular processes (i.e., hardening of each slip system) are likely important throughout the postseismic interval but certainly dominate the very earliest deformation that must proceed for precedes the transfer of stresses among grains to ensue.”
86: I suggest that, for a broader audience, you should explain briefly but clearly what GNDs are and why they are important.	Added
86-95: this last part of the introduction contains unexplained specialist terminology that detract somewhat from the message. Some explanations are given in the method, which helps, but this should be done even further so that all procedures and codes applied are well explained.	We have added brief explanations of the terminology in the Introduction (for ‘HR-EBSD’, ‘geometrically necessary dislocation’, residual stress’ and ‘long-range internal stress’) and the Methods (for ‘autocorrelation’). We have also added several new passages to the Methods section to explain the procedures more fully.

Table 1 and 2 In Methods are useful to gather an immediate idea of the datasets available. They should be referred to as soon as relevant in the results.	Added
Figure 1. – A summary such as this: “Fig. 1 is a visual representation of GND density from HAR-EBSD showing results from a variety of deformed and undeformed sample. All low T samples were deformed at high Pc, while all high T samples were deformed at low Pc. All samples, except single crystals, have grain size <10um. Only 2 high T aggregate samples were deformed at ~20% strain and faster strain rates of $\sim 10^{-4}$ /s.” in the caption or in the text would help the reader get to grips with the variety of conditions investigated.	Added
I assume the HAR-EBSD data comes from the area of each individual map in figure 1. One immediate observation is that each map is of a different size and contains considerably different number of grains (for aggregates) and therefore of boundaries too. My concern therefore is that the Authors have not considered how representative of the whole sample microstructure each individual area is. As additional work, an assessment of the optimal representative area for each sample microstructure should be carried out. This can be done by testing the area threshold by which, above a minimum area size map, the sample characteristics measured remain the same. This investigation is important because it will reduce errors in Figures 3, 4 and possibly 5, will reveal more reliable trends of stress probability distribution between different temperature experiments and will therefore result in more reliable interpretations of sound datasets.	We have added detailed new analysis (Figure 3), as suggested, which demonstrates that the mapped areas are of sufficient size to reliably characterise the stress distributions.
From qualitative observation it is clear in Figure 1 that high Pc, low T deformed samples generally have grains with higher GND density. Has any TEM work been done to document the actual defect content of these grains and compare it with the low Pc, high T deformed samples? TEM would identify dislocations as well as more complex defects, stacking faults, micro-deformation bands, microcracks. It seems important to have structural evidence to back the statement that dislocations interactions maybe the same in the low T and high T samples.	We have recently published detailed TEM analyses of the low-temperature samples (Wallis et al., 2020, EPSL), which can be compared to the abundant literature on samples deformed at high temperatures (e.g., Green and Radcliffe, 1972; Phakey et al., 1972). However, the details of the dislocations are less relevant to the overarching topic of the present manuscript, as we explain in the following text, which we have added to the Discussion: “Although there can be differences in the types, densities, and/or distributions of dislocations generated at low and high temperatures ^{30,33,34,36–38,50} , the stress fields of individual dislocations have negligible temperature dependence (only that of the shear modulus). Importantly, the new results (Figures 1, 2, and 4–6) demonstrate that there can be close similarity in the net stress fields of the populations of dislocations generated in each temperature regime. Therefore, there is similar potential for long-range interactions among the ‘free’ dislocations within cell or subgrain interiors.”
An additional single crystal deformed at high T would have been a good measure for comparison with low T single crystal and high and low T aggregates.	We have previously published detailed analyses of single crystals deformed at high temperatures (Wallis et al., 2017, JGR) and we draw these into the Discussion early on.
The layout of both Figures 1 and 2 may be improved by adding information on HIPing T for b) and by organising samples by T (e.g. all low T on the left, all high T on the right). Also, in Figure 2 the scale label is unclear, until later on, when one reads in the Methods that residual stress normalisation was done by subtracting the grain mean stress to σ_2 . I would encourage the Authors to ensure, within space limits, that all figures and concepts are clear and self-sufficient on first read.	Added
137: “the probability distribution of...” add “the normalised residual stress”.	Added
Figure 3: I found it informative (although subject to how representative of each sample each area analysed is, particularly for the deformed aggregates), but I would like to learn which sample is which directly from the figure, without having to try and double guess. You could add this information in the legend.	Added
Explain in the caption why the deformed single crystal peak is not centred on 0 GPa like all other peaks.	Added
Something I have been thinking about for a little while is the fact that, at least in my experience, residual stresses measured using HAR-EBSD on a variety of materials deformed at a variety of conditions (e.g. calcite, feldspar, olivine in this study and possibly other materials), seem to be of the order of between 0.5 and 1 GPa or thereabout. I would be interested, out of curiosity, on the Author’s comments on this and the potential reasons, given their experience with the application of HAR-EBSD to deformed samples that display large rotations.	This is simply because we have been analysing materials with broadly similar elastic moduli and dislocation densities that were often deformed under broadly similar homologous temperatures. The measured stresses are not ‘phantom strain’ artefacts generated by large lattice rotations as they are also present in samples with only small rotations, such as the single crystal San382t, which exhibits total rotations of only $\sim 1^\circ$ within the HR-EBSD map area. Another example is provided by the nanoindents in olivine presented by Wallis et al. (2020, EPSL), around which the stress fields of the

	indents extend beyond the zone of lattice rotation and into otherwise undeformed crystal.
142: Note that Figure 4 shows the cumulative probability distribution of normalised residual stresses, not the normal probability distribution. Correct text and caption.	Clarified
In the same line, distributions are shown for each area analysed, not for each sample. Correct this too.	Removed
I find Figure 4 rather confusing for both, how it is represented and how it is explained. It is difficult to pick trends and what they mean/fit with the interpretations. The restricted second moments need to be explained in a way that is accessible to the broader audience. This is true for the Method part at lines 348 onwards too. My advice to the authors here would be that they reflect on the key message they want to convey with these diagrams and redraft diagrams and text to achieve maximum clarity on this.	We have added three new passages of text to the Methods and modified the figure caption to make these plots clearer.
I have a similar comment for the use of the autocorrelation function in Figure 5: it needs to be explained more clearly (how it is obtained, where is it taken, what does it show). Understanding how the “peak” is defined is important to understanding how the correlation length of the stress field is determined.	Added
196-199: it might be that I am missing some key piece of information, but I find it puzzling that the high Pc, low T samples display a high number of grains with high GND density (considerably higher than high T samples) and yet the residual stresses measured in high T and low T samples are comparable. The authors say that this demonstrates “the high stress magnitudes recorded by the samples deformed at high temperatures are also predominantly imparted by the dislocation content”. Could the Authors explain why such different GND densities should be associated with similarly high residual elastic stresses? 211-213: but then perhaps the same GND density differences don’t seem to explain the similarity in the magnitude of stress heterogeneity between low T and high T aggregates...	Only one of the samples (San372b) deformed at low temperature has higher GND density than the samples deformed at high temperatures (Figure 1) and that sample does exhibit greater stress heterogeneity (the broadest peak in Figure 2 and the steepest gradient in Figure 4b), so there is no conflict here.
205: check reference cited here as it does not seem to be directly relevant to the statement made as it is concerned with the study of single crystals.	Added Hirth and Kohlstedt (2015)
225: need to add a reference after “...temperatures.”	Added Hansen et al. (2019)
261-end: Mussi et al. 2017 and mat sci references such as Koslowski et al., 2002 are missing.	Mussi et al. (2017) focussed on short-range, rather than long-range, dislocation interactions so isn’t relevant here. However, we have added that reference to the Introduction.
Reviewer 3	
I wonder: “Is testing the hypothesis enough for publication?” In their last sentence of the "introduction", the authors write: “If the hypothesis is supported by our results, these observations will provide the basis for a new generation of rheological models of transient creep, rooted in the microphysics of intracrystalline deformation.” To stop after validation of the hypothesis occurs to me like what one would do when writing a proposal. In a publication, I would like to learn at least how the models would look like and what would change in our perspective of Earth processes once they are at hand? Better even, the authors would formulate a model (based on their observations) and test its predictions against independent observations. To me, however, the manuscript is rather vague about what observations exactly shall be modeled.	We maintain that a novel, high-quality test of an important hypothesis is enough to warrant publication in a high-impact journal. Some of the information that the reviewer asks for was already included in the final paragraph of the Discussion. However, we have expanded that paragraph with the following text to provide a fuller perspective: “The form of a new model for transient creep should exploit these new constraints on the underlying processes by incorporating a system of equations derived from the microphysics of dislocation glide, recovery, and/or the evolution of back stress. A model for transient creep based on these intragranular processes could be compared to experimental data from aggregates to test whether additional processes, such as grain interactions ²⁹ , contribute significant effects. Ultimately, the development of a model for transient creep that can be explicitly related to specific key microphysical processes will provide more robust estimates of the evolution of mantle viscosity over the earthquake cycle. Moreover, by identifying the characteristics of stresses heterogeneity (i.e., the form of the probability distributions, typical length-scales and spatial distributions) in experimental samples deformed at high temperatures, we provide a new set of criteria against which to compare natural rocks to test the relevance of associated models of transient creep to the upper mantle.” The full development and testing of the new model for transient creep is far beyond the scope of a single paper (our group is currently preparing four manuscripts on these topics) but the present manuscript represents a step change in our understanding of the key processes to focus on.

The presentational aspects of the manuscript need further attention, as for example clarifying what is presented in the figures. Please find attached a digitally annotated pdf-version of the manuscript addressing problems with wording etc.	We have worked through the annotated manuscript and adopted the majority of the reviewer's suggestions. Most of the suggestions that we did not adopt stemmed from apparent misunderstanding about the method of measuring stress, which we hope is clearer in the revised manuscript due to the expanded Methods section.
my concerns about the approach: The authors try to deduce two quantities from EBDS-patterns 1) densities of (geometrically necessary; nominally sessile) dislocations 2) stress One may fundamentally question whether it is possible to derive "independently" two different physical quantities from a "single" measurement; I am not saying it is impossible but one had to demonstrate to what extent they are indeed independent or to what extent they exhibit correlations due to the method. To some extent, I am not so sure whether testing the hypothesis actually requires determining both, dislocation density and stress. The main objective, to my understanding, is to demonstrate the similarity of the deformation structures of samples deformed at different temperatures.	We get the impression that the Reviewer has missed our references to our recent review paper on HR-EBSD and its application to geological materials (Wallis et al., 2019, JGR), which deals with all the detailed methodological points raised. Therefore we have added explicit references to that paper in the Methods section and have also added substantial new text summarising the main points of the technique. We emphasise that this is not a single measurement. The elastic strains (with associated stresses) and lattice rotations (with associated GND densities) are definitely mathematically independent and can be measured fully independently. We can measure stress where there are no GNDs and vice versa. We have added description of this procedure.
My biggest problem relates to the method used to derive strain and stress of individual grains. The methods section is not very clear about the chosen procedure. Actually, looking at the referred earlier publication of the first author provided me with: "Stored diffraction patterns were reanalyzed using the HR-EBSD approach of Wilkinson et al. (2006) and Britton and Wilkinson (2011, 2012a). A reference pattern with high band contrast was chosen from each map. One hundred regions of interest (ROIs) were selected from each diffraction pattern and cross correlated with the corresponding ROIs in the reference pattern to determine the relative shift of each ROI. A displacement gradient tensor was fitted to the shifts in each pattern, allowing calculation of the lattice rotations and elastic strains relative to the reference pattern, both with a sensitivity of approximately 10^{-4} (Wilkinson et al., 2006). Residual stress variations were calculated from the elastic strain variations using the elastic moduli of olivine at 1 atm and 273 K (Abramson et al., 1997). In this analysis, the stress component normal to the sectioned specimen surface is assumed to be relaxed to zero. Strains are calculated in the reference frame of the microscope stage, but we present components of the stress tensor in the crystal reference frame to aid interpretation of their impact on dislocation processes." Even this description did not dispel my doubts. Microstructural patterns can only be used to derive "spatial variations in strain" and from them "spatial variations in stress", but neither "strain" nor "stress".  • The first step is to derive a displacement-gradient matrix that then has to be separated into its symmetric and anti-symmetric parts. • The next step is to calculate the spatial variations of these, from which "spatial variations in strain" can be deduced by "gradient" operations. • The final step is to transfer "spatial variations in strain" to "spatial variations in stress" using a rheological model. From the description of the method, it seems that the authors did not do the second step of calculating the spatial variations of the symmetric and anti-symmetric part but treated the symmetric part of the displacement-gradient matrix as "strain" and transferred that to "stress". The simple example of a twin shows that this "abbreviated" method is wrong. One half of a twin has a huge strain compared to the other and the abbreviated method would infer an according stress though the twinned crystals is free of internal stresses for the most part. For the twin, spatial CHANGES in strain occur only at the twin boundary and the field analysis of changes will correctly reflect the microphysics, from which we know that the remaining misfits of	Unfortunately, the reviewer missed the references to our recent review paper, which explains the method in full detail (unlike the paper that he mentions) and would alleviate these doubts. Therefore, we have added a sentence that explicitly signposts the review paper and have also significantly expanded the associated summary of the stress measurements within the Methods section. For convenience, we also briefly respond to the reviewer's comments here. We do indeed calculate the displacement-gradient tensor and separate the (anti-)symmetric parts. However, the key point (that was in the original manuscript) is that these measured values are relative to the strain state of the lattice at the chosen reference point within each grain. In other words, the measurement automatically provides the spatial gradients in strain and therefore the additional calculation that the reviewer suggests isn't required. Then we do indeed convert the strains to stresses using Hooke's law.

atomic layers at twin boundaries cause only local internal stresses [reflected by the finite interface energy].	
1) What is the effect of the numerical calculations of the spatial fields of interest? Deducing (changes in) strain from EBSD-patterns requires numerical calculation of spatial derivatives from digital data. The correct method actually requires second derivatives. Any numerical calculation of derivatives involves loss of information that results in “smoothing”. It is thus critical to examine the limits of interpretation one can apply to the spatial characteristics in the light of the numerical procedure.	Following on from the previous point, we do not need to calculate spatial derivatives of the elastic strains as we are already measuring how the strain state at each point differs from that at the reference point. However, we do take the spatial derivatives of the lattice rotations (i.e., gradients in plastic strain) to compute GND densities and we have added comments on this noise to the Methods section.
2) What are the limits of the inherently two-dimensional analysis? The surface of the samples is stress-free at the point of measurement; no information on vertical displacement available.	We have added the following paragraph to the Methods section to clarify: “HR-EBSD data are acquired on two-dimensional free surfaces of the specimens, which imposes constraints on the measurements of GND density and stress heterogeneity that can be obtained. Gradients in lattice orientation in the direction normal to the specimen surface cannot be observed and therefore dislocations that generate orientation gradients only in this direction cannot be detected^{40,48}. Therefore, the measured GND densities represent lower bounds on the total GND content. The presence of a free surface also relaxes the normal stress acting on that surface, while the other stress components are modified to a lesser extent due to the Poisson effect and changes in the tractions on the surface^{40,64,65}. Here, we focus on the σ_{12} component as this component is least modified by sectioning the samples and generally most closely pertains to the glide forces on dislocations during deformation.”
3) What happens to the samples during quenching? What happens to glissile dislocations? How do they affect the structure of sessile dislocations?)	We have added the following sentences to clarify: [regarding the low-temperature experiments] “We also note that these samples do not exhibit any reverse plastic strain during the final unloading³¹, which suggests that the dislocation microstructures prior to unloading are preserved in the final sample material.” [regarding the high-temperature experiments] “At the end of these experiments, the samples were quenched to below 800°C in less than 300 s while the maintaining the final load. This procedure has been demonstrated to preserve the steady-state dislocation microstructure during quenching⁵⁰.”
4) What is the effect of the convolution of the spatial characteristics of the real misorientation pattern and the resolution (size) of the electron beam? How does “filtering out”/“lack of indexing” for individual spots affect the outcome?	The spot size of the electron beam ($\ll 1 \mu\text{m}$) is much smaller than the typical length scale ($> 1 \mu\text{m}$) of lattice orientation gradients (i.e., misorientation, manifested as heterogeneous GND densities in Figure 1) and therefore has negligible impact on their appearance. To go one step beyond the reviewer’s question (for completeness), the finite area of the specimen surface that generates the diffraction pattern does limit our ability to detect the very highest stresses adjacent to dislocation cores due to some averaging over the surrounding (lower-stress) area. This point was already mentioned in the manuscript as “Several of the curves depart from straight lines at the highest stresses due to averaging of the elastic strains over the finite volume illuminated by the electron beam⁴⁵.”
[Referring to the Introduction where we state “We analyse the stress distributions in terms of the theory, established in the materials sciences, for stress fields of a population of dislocations to test the causality between stress heterogeneity and the dislocation content.”] I am lost: you do not have independent measures for the two but only EBSD maps. Since you use the “same” information to deduce the two fields I am not sure you can make this check.	This comment appears to stem from a misunderstanding about our method. We are not deriving our dislocation densities and stresses from processing the same (ordinary) EBSD data. We are using an entirely different method of analysing the diffraction patterns to map the symmetric and antisymmetric parts of the displacement-gradient tensor, which are mathematically and physically independent and can be used to calculate stresses and GND densities respectively. We have clarified this point by expanding the Methods section extensively, in particular including the following comments: “The displacement-gradient tensor was determined for each diffraction pattern by fitting the shifts measured by cross correlation. The symmetric and antisymmetric parts of the displacement-gradient tensor describes elastic strains and lattice rotations, respectively. As such, the elastic strains (with associated stresses) and lattice rotations (with associated GND densities) are mathematically independent and do not necessarily occur simultaneously in the material or HR-EBSD data.”
[Regarding referring to figures by opening paragraphs with “Figure X presents...”] Reviewer 3 suggested updating the text to refer to figures parenthetically.	As the figures report the essential data that forms the backbone of the work and must be inspected by all readers, we consider it more appropriate to refer to them actively at the beginning of the sentences, rather than making assertions in the sentences then referring to the figures parenthetically as an afterthought.

REVIEWER COMMENTS

Reviewer #1 (Remarks to the Author):

The authors have thoughtfully addressed the comments brought up in review, added more helpful description to their methods, results (including new figures) and interpretation, and clarified their arguments on how these new data can be used to develop new models for transient creep in the mantle. I am happy to recommend publication.

Greg Hirth

Reviewer #2 (Remarks to the Author):

Dear Authors,

I am satisfied that sufficient improvements have been made to the manuscript in response to my comments. In particular I appreciate the extensive additions made to the Methods section to explain the use of HR-EBSD techniques. This is helpful for readers of Nat Comm articles.

I also note the additional diagrams produced to demonstrate the validity of the data in terms of representative area size for corresponding microstructure.

I therefore have no further comments and consider this manuscript fit for publication.

With best wishes

Betty Mariani

Reviewer #3 (Remarks to the Author):

You asked me to review a revised version of “Dislocation interactions in olivine control postseismic creep of the upper mantle” authored by David Wallis, Lars N. Hansen, Angus J. Wilkinson, and Ricardo A. Lebensohn. The authors obviously took the various comments by all reviewers serious and I am grateful for their clarifications. My concerns on the significance of the spatial scales, on which observations are reported, and of observations in the light of the phrased hypothesis as well as of the hypothesis itself remain. To me, the significance of the observations remains unclear. Dislocations interact with each other and interfaces (grain boundaries), that is what they do. Microstructures will “qualitatively” reflect these interactions document, but I see little chance for deducing their quantitative relevance for rheological relations without actual experiments.

Nevertheless, I do not want to be an obstacle and consider this report more like “providing the authors with a curious reader’s perception of their manuscript”. The attached pdf-version of the manuscript has a few comments in addition to the ones below but mostly contains “highlightings” that helped me keep track of the manuscripts contents.

Kind regards,

Joerg

size issues

Below, I collect the various statements regarding spatial scales of measurements and observations. (Maybe the “technical aspects of scales” could be collected in one concise paragraph?) If I did not overlook this information, the authors still do not seem to report (i) the beam size that is obviously critical for the scale on which information can retrieved, (ii) how much of an individual grain is actually captured (spots on grain boundaries and fraction of “failed” measurements per grain?), and (iii) how the gradient is calculated, i.e., over which length scale it is averaged.

to (ii): The authors report “As the full datasets contain between approximately 2×10^4 and 4.8×10^5 measurements” but do not say much about the spatial distribution of these measurements.

to (iii): from the rebuttal: We do indeed calculate the displacement-gradient tensor and separate the (anti-)symmetric parts. However, the key point (that was in the original manuscript) is that these measured values are relative to the strain state of the lattice at the chosen reference point within each grain. In other words, the measurement **automatically provides the spatial gradients** in strain ...

My apologies for my notoriety (and potential ignorance): I do not understand what the authors mean by “automatically”; after determining a difference between two values, i.e., one at a point of interest and one at a reference point one still has to “divide” by a length scale to get a gradient.

The list below documents that “everything investigated and observed” seems to be on the scale of 1 μm . Furthermore, I infer that the “resolution” of the method is indicated by the “digits” of the autocorrelation that in turn is identical with the step size (by chance?).

abstract: ... stress heterogeneities of ~ 1 GPa that are imparted by dislocations and have correlation lengths of ~ 1 μm .

introduction: ... all other samples are aggregates with grain sizes < 10 μm

qualitative only: ... lattice distortion by using cross correlation to track shifts in subregions within diffraction patterns ...

results: ... the grains in these samples contain irregular-shaped patches, a few micrometres across ...

... case, stress typically varies smoothly between domains of stress on the order of 1 μm across, with alternating sign and magnitudes again on the order of 1 GPa.

line 197: Several of the curves depart from straight lines at the highest stresses due to averaging of the elastic strains over the finite volume illuminated by the electron beam⁴⁵.

Figure 6: autocorrelation is shown with a “resolution” of 0.1 to 0.2 μm

Table 2 (methods section) step size 0.15/0.2 μm for aggregates, 0.5 μm for single crystal

line 464: However, for the majority of grain orientations, HR-EBSD can resolve GND densities down to a noise level on the order of 10^{13} m^{-2} at the step sizes of 0.15–0.5 μm employed in this study (Table 2)48. This spatial resolution allows local intragranular regions of elevated GND density to be resolved, whereas regions of low GND density may be obscured by noise.

Another technical aspects that I did not grasp: Why is the “noise floor” different for single crystals and for aggregates, i.e, $1\text{E}12$ vs. $3\text{E}13 \text{ m}^{-2}$.

“indirect” reasoning:

line 86: “Specifically, ...”

line 109: “... hypothesis that long-range dislocation interactions contribute to strain hardening of olivine aggregates at high temperatures is supported by our results ...”

Here, I would “boldly” comment that the presence of long-range interactions is neither surprising (to me) nor does it provide evidence for hardening in the absence of mechanical data. To me, the approach cannot replace doing the respective experiments. All in all, I wonder why the authors do not do the same comparison of tests on singles crystals and aggregates as the one described for low temperature.

“differences”

The collection of citations from the manuscript below shall indicate that I get confused about when authors consider observations similar or different and how that relates to their aim to test the hypothesis.

effect of temperature

Microstructural **differences** of samples deformed at different temperatures:

line 127: “The samples deformed at high temperatures, PI-1488, PI-1523, and PI-1519, contain GND densities broadly comparable to those of San382b [deformed at room temperature] but typically exhibit smoother variations in GND density within grains and more linear arrays of GNDs (e.g., PI-1523) than the aggregates deformed at room temperature.”

The probability distributions of normalised σ_{12} in the aggregates deformed at low temperatures are similar to, or broader than, those of aggregates deformed at high temperatures.

line 248: The stress distributions of samples deformed at both low and high temperatures exhibit high stress tails that deviate from normal distributions (Figures 4a and 5) and are typical of materials deformed by dislocation-mediated mechanisms, even at low temperatures^{44-47,49}.

Another “bold” note: I find “dislocation-mediated mechanisms” too unspecific in the context of this work that specifically aims to show that a specific dislocation interaction is more relevant than others.

effect of sample type

differences among single crystals and aggregates:

line 145: “Sample San382t [single crystal] contains bands of elevated stress of alternating sign that vary in magnitude on the order of 1 GPa over distances of a few micrometres. The deformed aggregates exhibit stress distributions that are qualitatively similar to each other but lack the ordered structure displayed by San382t [single crystal].”

comment: What do you mean by “each other”? If that refers to “aggregates irrespective of temperature at which they were deformed” the statement seems to contradict that on line 127.

“The stress distributions are broader in the deformed aggregates than in the deformed single crystal but all deformed samples contain distributions that extend beyond ± 1 GPa. The maximum absolute value of normalised σ_{12} in the deformed single crystal is 3.3 GPa, whereas those of the deformed aggregates are in the range 7.4–13.5 GPa. These maximum values are consistent with the yield stress of olivine at short length scales and room temperature³².”

discussion: line 237 “A surprising result of the present study is that, in contrast to single crystals, the aggregates of olivine deformed at 1150–1250°C exhibit stress heterogeneity with magnitudes again frequently on the order of 1 GPa, closely comparable to the aggregates deformed at room temperature (Figures 1, 2, 4, and 5).”

comment: Are the ranges in stresses relevant or is (as phrased in line 145) everything on the order of 1 GPa and therefore similar? What is the (quantitative) meaning of “short length scales”?

discussion: line 241 “One hypothesis for the cause of increased stresses in aggregates, relative to single crystals, deformed at high temperature is that they are imparted by thermal contraction.”

comment: Reorganization of dislocations (and dislocation arrangements) during quenching will be largely different in single crystals and polycrystalline aggregates due to the presence of the grain boundaries.

line 269: “As GNDs impart significant non-cancelling stress fields, differences in GND density explain the observed differences in the magnitudes of stress heterogeneity between single crystals and aggregates deformed at both low and high temperatures (Figures 1, 2, 4, and 5)³⁸.”

comment: What is the meaning of “non-cancelling”? Summed over an entire grain the stresses have to cancel, or?

what is the significance of residual stress?

discussion: line 251 "... indicates that the tails of the distributions follow $P(\sigma) \propto \sigma^{-3}$ in all the deformed samples, as expected of stress fields generated by dislocations⁴⁴⁻⁴⁷ (Methods)"

comment: So, the very observation is rather unspecific regarding the interaction mechanism?

line 254: "... provide strong evidence that the high stress magnitudes recorded by the samples deformed at high temperatures are also predominantly imparted by the dislocation content."

comment: What else could cause high residual stresses? Be clear about alternatives?

"The interpretation that dislocations are the dominant cause of stress heterogeneity in the aggregates deformed at high temperatures is consistent with the previous analysis of single crystals of olivine deformed at similar temperatures³⁸."

comment: May be I am (again) totally off but the method can provide heterogeneity of residual stress only for individual grains but not among grains of aggregates, so I consider "cause of stress heterogeneity in the aggregates" misleading.

"In those samples, the stress heterogeneity was clearly controlled by the dislocation content rather than constraints imposed by neighbouring grains (as is also the case for San382t37)."

comment: I do not understand, a single crystal does not have neighboring grains.

line 291: "conclusion" Therefore, long-range internal stresses are increasingly apparent as a ubiquitous characteristic of olivine deformed by dislocation-mediated mechanisms.

comment: The above statement is rather vague.

Reviewer's comment	Authors' response
To me, the significance of the observations remains unclear. Dislocations interact with each other and interfaces (grain boundaries), that is what they do. Microstructures will “qualitatively” reflect these interactions document, but I see little chance for deducing their quantitative relevance for rheological relations without actual experiments. line 109: “... hypothesis that long-range dislocation interactions contribute to strain hardening of olivine aggregates at high temperatures is supported by our results ...” Here, I would “boldly” comment that the presence of long-range interactions is neither surprising (to me) nor does it provide evidence for hardening in the absence of mechanical data. To me, the approach cannot replace doing the respective experiments. All in all, I wonder why the authors do not do the same comparison of tests on singles crystals and aggregates as the one described for low temperature.	Actually, it cannot be taken for granted that dislocations will generate the long-range elastic interactions that are the focus of the manuscript. We have added the following sentences to the Discussion to make this clear “This is by no means a foregone conclusion for at least three reasons. First, if the plastic-strain fields were typically near homogeneous, most dislocations would be statistically-stored dislocations and their long-range stress fields would largely cancel. Second, strain hardening can occur by short-range dislocation interactions (e.g., formation of junctions) that do not require long-range internal stress³³. Third, recovery mechanisms, such as subgrain-boundary formation and (sub)grain-boundary migration, can potentially reduce GND densities and long-range internal stresses within subgrain interiors.” The reviewer is correct that experiments are needed to quantitatively calibrate a new rheological model for transient creep, but that is not the purpose of this manuscript. We set out a very clear and important hypothesis with the aim of identifying the key microphysical processes that future experiments should target and that the associated equations should be based on. To elaborate on the implications of our results for quantitative models, we have added the following text to the Discussion: “Specifically, our microstructural observations imply that a quantitative model for high-temperature transient creep should include a back-stress term that is subtracted from the applied stress so that dislocation glide proceeds under the action of an effective stress³¹. With such a formulation, analogous to that employed for low-temperature plasticity by Hansen et al.³¹, changes in applied stress can result in negative effective stresses and therefore generate reverse flow. This viscoelastic behaviour is an important component of recent geodetic analyses of postseismic deformation^{1–4,15} and our microstructural observations suggest that it results, at least in part, from back stress generated by long-range dislocation interactions.” For context, we have already published the mechanical data and rheological model associated with the samples deformed at low temperature (Hansen et al., 2019, J.G.R.) and, in fact, we have just completed an initial set of experiments to calibrate the kinetics of the key processes identified in the present manuscript and the information from this manuscript is essential to that effort. A preprint is available here for further details: https://doi.org/10.1002/essoar.10504736.1. Likewise, we have previously published analyses of single crystals deformed at high temperatures (Wallis et al., 2017, J.G.R.) and already describe the relevance of those results in detail in the Discussion.
(Maybe the “technical aspects of scales” could be collected in one concise paragraph?) [The reviewer lists examples of the aspects that he is referring to, including step size of EBSD measurements, resolution of autocorrelation functions, correlation lengths of stress fields, and grain sizes] If I did not overlook this information, the authors still do not seem to report (i) the beam size that is obviously critical for the scale on which information can be retrieved, (ii) how much of an individual grain is actually captured (spots on grain boundaries and fraction of “failed” measurements per grain?), to (ii): The authors report “As the full datasets contain between approximately 2×10^4 and 4.8×10^5 measurements” but do not say much about the spatial distribution of these measurements.	We have added the requested information to the Methods sections, as follows. It is more appropriate to present the information in the relevant sections than in a single paragraph. (i) “The footprint of the source region of the diffraction patterns on the specimen surface is estimated to be < 100 nm across. Step sizes were in the range 0.15–0.5 μm, which are significantly smaller than the grain sizes of the samples (3–700 μm, Table 1) and are therefore suitable for resolving intragranular GND densities and stress heterogeneity.” “The spatial resolutions of the autocorrelation functions are equivalent to the step sizes used to acquire the EBSD data and are therefore in the range 0.15–0.5 μm (Table 2).” (ii) We have added two columns to Table 2 that report the fractions of pixels that were successfully included in the EBSD and HR-EBSD datasets. “Initial indexing rates were in the range 92–100% (Table 2). Nonindexed pixels were commonly located on grain boundaries. Nonindexed pixels with at least seven indexed neighbouring pixels within the same grain were filled with the average orientation of the neighbouring pixels.” “After the cross-correlation procedure, we filtered out results from the stress datasets for which the normalised peak in the cross-

	correlation function was < 0.3 and those with a mean angular error in the deformation gradient tensor $> 0.004^{43}$. The final stress datasets include at least 69% of the pixels in each map (Table 2). For datasets of GND density, we filtered out both those pixels that failed the quality criteria and their neighbouring pixels to remove GND densities that were calculated from potentially spurious orientation gradients involving the poor-quality pixels. Most of the pixels that failed the quality criteria were located in intragranular regions with large lattice rotations of at least several degrees relative to the lattice orientation at the reference point.”
(iii) how the gradient is calculated, i.e., over which length scale it is averaged. to (iii): from the rebuttal: We do indeed calculate the displacement-gradient tensor and separate the (anti-)symmetric parts. However, the key point (that was in the original manuscript) is that these measured values are relative to the strain state of the lattice at the chosen reference point within each grain. In other words, the measurement automatically provides the spatial gradients in strain ... My apologies for my notoriety (and potential ignorance): I do not understand what the authors mean by “automatically”; after determining a difference between two values, i.e., one at a point of interest and one at a reference point one still has to “divide” by a length scale to get a gradient.	Our measurements provide the difference in lattice orientation and elastic strain state relative to those at a reference point within each grain. The measurements are collected on a regular grid so we can divide those differences by the distances between points to get the spatial gradients. However, note that this last step is not required for the elastic strains and corresponding stresses. The measurements (antisymmetric part of the displacement-gradient tensor of the diffraction patterns) already provide us with a map of elastic strain heterogeneity. In contrast, we do divide the differences in lattice orientation (coming from the symmetric part of the displacement-gradient tensor of the diffraction patterns) by distance in the calculation of GND densities, which are the microstructural expression of gradients in plastic strain. Full mathematical descriptions are provided by Wallis et al. (2016, Ultramicroscopy; 2019, J.G.R.), which is referenced in the manuscript.
Another technical aspects that I did not grasp: Why is the “noise floor” different for single crystals and for aggregates, i.e., $1E12$ vs. $3E13$ m-2.	The noise floor depends on both crystal orientation and mapping step size, as outlined in the Methods.
line 127: “The samples deformed at high temperatures, PI-1488, PI-1523, and PI-1519, contain GND densities broadly comparable to those of San382b [deformed at room temperature] but typically exhibit smoother variations in GND density within grains and more linear arrays of GNDs (e.g., PI- 1523) than the aggregates deformed at room temperature.” The probability distributions of normalised σ_{12} in the aggregates deformed at low temperatures are similar to, or broader than, those of aggregates deformed at high temperatures.	There is no contradiction here. Both the GND densities and stress heterogeneities are similar among the samples deformed at low temperature and those deformed at high temperatures. Any differences in the distributions among these samples are minor compared to the differences with the single crystal or undeformed samples.
line 248: The stress distributions of samples deformed at both low and high temperatures exhibit high stress tails that deviate from normal distributions (Figures 4a and 5) and are typical of materials deformed by dislocation-mediated mechanisms, even at low temperatures^{44–47,49}. Another “bold” note: I find “dislocation-mediated mechanisms” too unspecific in the context of this work that specifically aims to show that a specific dislocation interaction is more relevant than others.	We have deleted this phrase.
line 145: “Sample San382t [single crystal] contains bands of elevated stress of alternating sign that vary in magnitude on the order of 1 GPa over distances of a few micrometres. The deformed aggregates exhibit stress distributions that are qualitatively similar to each other but lack the ordered structure displayed by San382t [single crystal].” comment: What do you mean by “each other”? If that refers to “aggregates irrespective of temperature at which they were deformed” the statement seems to contradict that on line 127.	We have reworded this sentence to the following to clarify that we are comparing samples and commenting on the spatial distributions of stress heterogeneity (not the magnitudes of the stresses): “The deformed aggregates, irrespective of the temperature at which they were deformed, exhibit spatial distributions of stress heterogeneity that are qualitatively similar to each other and lack the ordered structure displayed by San382t.”
“The stress distributions are broader in the deformed aggregates than in the deformed single crystal but all deformed samples contain distributions that extend beyond ± 1 GPa. The maximum absolute value of normalised σ_{12} in the deformed single crystal is 3.3 GPa, whereas those of the deformed aggregates are in the range 7.4–13.5 GPa. These maximum values are consistent with the yield stress of olivine at short length scales and room temperature³².” discussion: line 237 “A surprising result of the present study is that, in contrast to single crystals, the aggregates of olivine deformed at 1150–1250°C exhibit stress heterogeneity with magnitudes again frequently on the order of 1 GPa, closely comparable to the aggregates deformed at room temperature (Figures 1, 2, 4, and 5).”	The overarching characteristics of the stress distributions are similar among all the deformed aggregates as evident in Figures 1–5 and, specifically, the full widths at half maximum are all on the order of 1 GPa (Figure 3). The range of 7.4–13.5 GPa refers to the few measurements at the extremes of the distributions and are relevant in demonstrating that the olivine is not supporting stresses that are greater than its yield stress (as requested by a different reviewer in the last round of revisions). To clarify, we have added “(e.g., full widths at half maximum, Figure 3)” at the relevant point in the discussion. We have replaced “short length scales” with length scales on the order of $1 \mu\text{m}$”.

comment: Are the ranges in stresses relevant or is (as phrased in line 145) everything on the order of 1 GPa and therefore similar? What is the (quantitative) meaning of “short length scales”?	
discussion: line 241 “One hypothesis for the cause of increased stresses in aggregates, relative to single crystals, deformed at high temperature is that they are imparted by thermal contraction.” comment: Reorganization of dislocations (and dislocation arrangements) during quenching will be largely different in single crystals and polycrystalline aggregates due to the presence of the grain boundaries.	We cool the samples under load and at rates that are well established to be fast enough to prevent significant modification of the dislocations arrangements and densities (in both single crystals and aggregates). We added the following sentence to the Methods section during the last round of revisions to clarify this point: “At the end of these experiments, the samples were quenched to below 800°C in less than 300 s while the maintaining the final load. This procedure has been demonstrated to preserve the steady-state dislocation microstructure during quenching⁵⁰.”
line 269: “As GNDs impart significant non-cancelling stress fields, differences in GND density explain the observed differences in the magnitudes of stress heterogeneity between single crystals and aggregates deformed at both low and high temperatures (Figures 1, 2, 4, and 5)³⁸.” comment: What is the meaning of “non-cancelling”? Summed over an entire grain the stresses have to cancel, or?	We have added the following sentence to clarify: “Unlike statistically stored dislocations, the stress fields of GNDs include a significant component that does not cancel over length scales greater than the average dislocation spacing.”
discussion: line 251 “... indicates that the tails of the distributions follow $P(\sigma) \propto \sigma^{-3}$ in all the deformed samples, as expected of stress fields generated by dislocations⁴⁴⁻⁴⁷ (Methods)” comment: So, the very observation is rather unspecific regarding the interaction mechanism?	At this point in the text, we are simply demonstrating that the stress fields are generated by dislocations and are not commenting on the interaction mechanism. Nonetheless, it is worth mentioning here that it is very unlikely that the inverse cubed dependence would arise by any other mechanism. Moreover, in the wider discussion, the form of the probability distributions is only one of several observations that collectively indicate the interaction mechanism. The other observations are the length scale of stress heterogeneity, the spatial correlation between high stresses and high GND densities, and the mechanical data from the low-temperature experiments.
line 254: “... provide strong evidence that the high stress magnitudes recorded by the samples deformed at high temperatures are also predominantly imparted by the dislocation content.” comment: What else could cause high residual stresses? Be clear about alternatives?	We have added the following sentence to clarify: “The alternative causes of residual stress, that grains interact by mutual exertion of forces on one another due to heterogeneous deformation, thermal contraction, and/or decompression, cannot generate this combination of characteristics.”
“The interpretation that dislocations are the dominant cause of stress heterogeneity in the aggregates deformed at high temperatures is consistent with the previous analysis of single crystals of olivine deformed at similar temperatures³⁸.” comment: May be I am (again) totally off but the method can provide heterogeneity of residual stress only for individual grains but not among grains of aggregates, so I consider “cause of stress heterogeneity in the aggregates” misleading.	The method can only measure intragranular stress heterogeneity and therefore cannot compare the absolute magnitudes of stresses between different grains. However, we can still measure intragranular stress heterogeneity for each grain in an aggregate and, importantly, stresses due to intergranular interactions (i.e., one grain pushing on another) would manifest as intragranular stress concentrations that we are able to detect. This effect is, in fact, evident in the stress data from that sample that has been isostatically hot pressed but is otherwise undeformed. Therefore, we can decipher the causes of stress heterogeneity in aggregates as well as single crystals. We have added the following sentences to the Methods to clarify: “As a different reference point is chosen within each grain, the absolute magnitudes of the stresses may differ by an unknown amount between grains. Nonetheless, the measurements are still sensitive to stresses arising from grain interactions in an aggregate of grains (i.e., the forces of grains indenting their neighbours), which will manifest as intragranular stress concentrations, as well as the intragranular stress fields of dislocations.”
“In those samples, the stress heterogeneity was clearly controlled by the dislocation content rather than constraints imposed by neighbouring grains (as is also the case for San382t37).” comment: I do not understand, a single crystal does not have neighboring grains.	We have replaced this sentence with the following to clarify: “During experiments on single crystals there are no neighbouring grains to generate stresses from grain interactions, and therefore any stress heterogeneity must result from the dislocation content”
line 291: “conclusion” Therefore, long-range internal stresses are increasingly apparent as a ubiquitous characteristic of olivine deformed by dislocation-mediated mechanisms. comment: The above statement is rather vague.	We have replaced this sentence with “Therefore, long-range internal stresses appear to be common in deformed olivine, even among samples deformed over a wide range of temperatures.”